# Farmer vs. Industrial Practices: Impact of Variety, Cropping System and Process on the Quality of Durum Wheat Grains and Final Products

**DOI:** 10.3390/foods12051093

**Published:** 2023-03-03

**Authors:** Marie-Françoise Samson, Anaïs Boury-Esnault, Ewen Menguy, Valentin Avit, Elodie Canaguier, Bruno Bernazeau, Patrice Lavene, Yuna Chiffoleau, Gregori Akermann, Kristel Moinet, Dominique Desclaux

**Affiliations:** 1IATE, Univ Montpellier, INRAE, Institut Agro, 34060 Montpellier, France; 2INRAE, UE DiaScope, UE 0398, 34130 Mauguio, France; 3INNOVATION, Univ Montpellier, CIRAD, INRAE, Institut Agro, 34060 Montpellier, France; 4Biocivam 11, 11800 Trèbes, France

**Keywords:** participatory research, ancient variety, gluten, farming system, pasta making process, protein, digestibility, semolina

## Abstract

The consumption of artisanal and organic pasta made on-farm from ancient varieties is increasing in France. Some people, namely, those suffering from digestive disorders following the consumption of industrial pasta, consider these artisanal pasta to be more digestible. Most of them have linked these digestive disorders to the ingestion of gluten. We analyzed in this study the impact of industrial and artisanal practices on the protein quality of durum wheat products. The varieties recommended by the industry (IND) were compared to those used by farmers (FAR): the FAR being on average much richer in protein. However, the solubility of these proteins analyzed by Size Exclusion-High Performance Liquid Chromatography (SE-HPLC) and their in vitro proteolysis by digestive enzymes vary little between the two groups of varieties, while differences between varieties in each group are observable. The location of grain production and the tested cropping systems (zero vs. low input) have a low impact on protein quality. Yet, more contrasting modalities should be studied to validate this point. The type of production process (artisanal vs. industrial) is, among those studied, the factor having the greatest impact on protein compositionPasta produced by the artisanal method contains a higher sodium dodecyl sulfate (SDS)-soluble protein fraction and are more in-vitro proteolyzed. Whether these criteria are indicative of what happens during a consumer’s digestion remains to be determined. It also remains to be assessed which key stages of the process have the greatest influence on protein quality.

## 1. Introduction

Durum wheat, *Triticum turgidum* L. ssp. *Durum*, is one of the few cereals intended exclusively for human consumption. It is generally transformed into semolina and then into pasta or couscous. Twenty years of participatory research carried out in the south of France with partners from farming, artisanal and industrial sectors have highlighted differences in practices between these different operators [1]. These differences are to be found not only in the choice of durum wheat varieties and the agronomic practices, but also in pasta processing. Farmers and craftsmen use mainly ancient (heritage) durum wheat varieties or those resulting from participatory breeding programs and grow them under organic conditions. They transform their harvest into semolina on a stone-mill and produce pasta under slow conditions by avoiding high-temperature drying. On the other hand, industrialists use mainly modern (elite) durum wheat varieties, grown in a conventional mode, ground on roller-mills and transformed into pasta under controlled conditions with high or very high temperature drying step [2].

Durum wheat grains contain about 14% protein [3]. Wheat proteins are classified according to their solubility [4]: albumins (soluble in water), globulins (soluble in saline solutions), gliadins (soluble in dilute alcohols) and glutenins (soluble in dilute acids and bases). Gliadins and glutenins are storage proteins. Gliadins are monomeric proteins, grouped into four classes: ω-, α-, β- and γ-gliadins. Glutenins appear in the form of polymers composed of two types of subunits: low-molecular-weight glutenin subunits (LMW-GS) and high-molecular-weight glutenin subunits (HMW-GS). During the transformation processes, under the effect of hydration and energy supply, these polymers rearrange, disulfide bonds disrupt and new bonds are created [5]. Gliadins interact with this glutenin skeleton by non-covalent bonds and together form gluten. Gliadins contribute to the viscosity and extensibility of gluten while glutenins are the major element of elasticity [6].

There is a current craze for local varieties or so-called “old” or heritage varieties. One of the reasons for this enthusiasm is the association made between local, regional products and attributes such as fresh, tasty, nutritional value, healthy and safe [7]. The selection of durum wheat, during the second half of the 20th century, resulted in an increase in yield and resistance to diseases, a decrease in protein content and a change in protein composition with the introduction of glutenin alleles favorable to pasta quality (i.e., HMW 7 + 8. LMW-2) [8,9] but also by a reduction in genetic diversity [10]. The proportions of the different protein classes have also changed; some authors [11,12] indicate that modern genotypes have a higher glutenin to gliadin ratio (Glu/Glia) than that of old varieties and that the expression of type B LMW subunits is twice as high in the former. At the same time, they noted no significant differences in the expression of α-, β- and γ-gliadins between the two groups, but they mentioned a significant decrease in ω-gliadins. In another article, De Santis et al. [13] reported a content of unextractable polymeric protein (UPP) twice as high for the modern variety Saragolla as for the old variety Cappelli and also a lower expression of ω-gliadins. Several studies comparing modern wheats and ancient wheats have attempted to link the harmful effects of gluten to the year of creation and the change in protein composition. It usually appears that old varieties are no less toxic than recent varieties [14,15,16]. However, the study by Ianiro et al. [17] showed that patients with NCGS (non-coeliac gluten sensitivity) expressed fewer gastrointestinal and extra-intestinal symptoms after ingesting pasta made with an “old” variety of durum wheat (cv.Senatore Cappelli, date of release = 1923) compared to ingesting pasta from a mixture of recent varieties. Some people suffering from digestive disorders and declaring themselves NCGS (non-coeliac gluten sensitive), questioned during a specific survey [18] or during personal discussions, declare that they can consume wheat-based farm products without experiencing the inconveniences they usually feel when eating industrial semolina or pasta.

From an agronomic point of view, the effects of nitrogen fertilization on the storage protein composition of cereals are well documented. A meta-analysis [19] carried out over the period 1960–2019 highlighted an increase in nitrogen fertilization from 9.8 to 93.8 kg N. ha^−1^. y^−1^. At the same time, this increase in fertilization resulted in a higher gluten content, with more gliadins. Over the period, the authors estimated the increase in the intake of gliadins in the ration, going from 2.4 to 3.8 kg.y^−1^, representing a jump of +58%. According to these same authors, this increase would be positively correlated with that of celiac disease. The increase in nitrogen fertilization resulted in an increase in the proportion of monomeric gliadins and, on the other hand, decreased the proportion of large glutenin polymers [20]. This result is supported by the data of Wan et al. [21] who indicated that increasing the level of nitrogen fertilization led to an increase in ω-gliadin expression with increased proportions of ω5 gliadin.

With regard to the processes, numerous works concerning their impact on the components of the finished product, in particular, proteins, have been carried out. The industrial process for making pasta differs from traditional methods by extensive refining of the semolina, control over its hydration, by controlling temperatures and pressures throughout the process and finally by the characteristics of drying (short duration and high or even ultra-high temperature). A high protein content (12 to 15% depending on the country) is required in the industry to produce semolina and give pasta good resistance to cooking, the absence of stickiness and optimal firmness [22,23,24]. When the protein content is lower, it can be compensated by strong gluten [25,26]. The improvement in the final quality of pasta has also been achieved through technological progress with the introduction of drying at high, or even very high temperature [27]. Drying at high temperature and especially very high temperature has made it possible to significantly improve pasta quality, in particular, when using medium-quality wheat [28]. At very high temperatures, the gluten network formed by gliadins and glutenins stiffens under the effect of heat via disulfide bonds, which results, on the biochemical level, in protein denaturation, reinforcement of the gluten around starch granules [29] and a decrease in protein solubility in many solvents such as acetic acid [30].

Protein digestibility is estimated to be over 90% in rich countries [31]. Food processes involving altering or modifying the supramolecular and molecular structures make them more accessible, or not accessible, to digestive enzymes [32]. Gluten proteins may be resistant to enzymatic hydrolysis due to the presence of proline-rich regions, and the process, depending on its intensity, may reduce digestibility [33,34], similarly to the presence of other compounds such as fiber [35]. There are few studies on the digestibility of proteins as a function of parameters such as the date of registration of the variety or the cropping system. Concerning soft wheat, Gulati et al. [36] note an increase in digestibility linked to the decrease in the protein content of “modern” wheat. Ma and Baik [37], after analyzing different species of wheat (soft wheat, durum wheat, einkorn, spelt, emmer), show that proteins that do not constitute gluten (albumins, globulins) affect protein digestibility.

This article is the result of a participatory research work involving farmers, actors in agricultural advice and the artisanal and industrial sectors, doctors, researchers and trainers from various disciplines, consumers and people diagnosed with NCGS. Their collaboration aimed to understand the combined impact of varieties, cropping systems, and processes on the quantity, quality and in vitro digestibility of gluten proteins in based-durum wheat products. They compared the protein quality of farmhouse and industrial semolina and pasta.

## 2. Materials and Methods

### 2.1. Grain Samples

Twenty varieties of durum wheat (*Triticum t. durum*) were grown and harvested at the DiaScope experimental station (INRAE, Mauguio, France 43°7′14″ N, 35°9′03″ E) in 2015–2016 (H2016 = harvest in 2016), 2016–2017 (H2017) and 2018–2019 (H2019) (No trial was conducted in 2017–2018). Among these varieties, eight categorized as IND have been chosen from the list of varieties recommended by the SIFPAF (Union of Industrial Pasta Manufacturers of France) and the CFSI (French Committee for Industrial Semolina); six varieties categorized as FAR are used by farmers and dedicated to the artisanal sectors of the Occitanie region; six other varieties, identified as DIV, are currently underutilized or no-longer-used genetic resources (Table 1).

The agronomic device was a randomized split plot with 3 replications implemented only under zero input (ZI) conditions in H2016 (previous crop: chickpea) and H2019 (previous alfalfa) and in two cropping systems in H2017: ZI (previous and antecedent Sainfoin, nitrogen residue: 80 units, no fertilizer, herbicide or pesticide input) and LI (low inputs) (previous chickpea, nitrogen residue: 80 units, supply of 150 units of Nitrogen + application of 2 herbicides). In H2019, varieties were also grown at the Purpan Engineering School (Toulouse, France 1°24′06″ E, 43°35′43″ N) under zero input conditions. At harvest, the grains were cleaned to remove all foreign elements and stored in a cold room.

### 2.2. Criteria Acquired on the Whole Grain

The thousand-kernel weight (TKW), the specific weight (SW) and the total protein content were measured on each sample of whole grain, by near-infrared spectroscopy (NIRS) using the calibrations developed by Foss (2017) and integrated in the Infratec Nova spectrometer (Foss, Hillerød, Denmark) located at INRAE-Mauguio. The principle behind NIRS is that specific organic molecules absorb specific wavelengths of near infrared light energy. The absorptions are directly correlated with the concentration of the organic molecules in the sample.

### 2.3. Milling Wheat Grains and Making Pasta

Three processes were carried out to produce semolina and pasta: “Artisanal 1”, “Artisanal 2” and “Industrial”.

Artisanal 1 process: ten varieties, harvested in 2016, were selected among the 20 varieties listed in Table 1 to make pasta on a farm in Bezouce (30320, France) by a farmer (François C.) who is used to producing durum wheat, grinding it, and making pasta on his own farm using an artisanal process. For the “Artisanal 1” process, the farmer has ground 5 kg of grain of each variety with his Tyrol-type stone-mill (SM 1). Then, he made tagliatelle on his P3-type pasta press (La Monferrina, Moncalieri, Italy), which has a mixing capacity of 3 kg and is capable of producing 8 to 10 kg.h^−1^. The pasta was then air dried (Table 2).

Artisanal 2 process: based on 2017 harvest, only two varieties of durum wheat belonging to “IND” group (cv. Anvergur, cv. Claudio) and two others belonging to “FAR” group (cv. LA1823 and cv. Bidi17) were used. The semolina was produced on a Tyrol-type stone-mill (SM 2) at the Lycée Agricole d’Auzeville (Castanet-Tolosan, France).

Subsequently, part of the semolina was transformed into artisanal pasta at the Lycée Agricole d’Auzeville on a pasta press Mac 30 (Italpast, Parma, Italy) equipped with a tagliatelle bronze die. The pasta was then dried on racks in a ventilated chamber at 37 °C.

Industrial process: the other part of the semolina (SM 2) was used to produce “industrial”-type pasta at JRU IATE (INRAE Montpellier, France) using an experimental mini-press (Sercom, Montpellier, France) from 700 g of semolina hydrated at 47% (dry basis). After hydration, the semolina was mixed for 20 min to form agglomerates which were then extruded under vacuum through a pappardelle die (width 33 mm, thickness approx. 0.6 mm), coated with Teflon and maintained at 40 °C. Pasta was then dried at very high temperature (90 °C) in a drying chamber regulated by temperature and humidity (CS-40, CTS, Auriol, France) (Table 2).

### 2.4. Preparation of Samples and Physico-Chemical Analysis

The grains and dry pasta were ground with a Cyclotec 1093 laboratory mill (Foss Tecator, Nanterre, France) for biochemical analysis. Pasta were cooked to their optimal cooking time (OCT) according to the AACC 66–50.01 standard [38], then drained and frozen before being freeze-dried and ground with an IKA A10 basic grinder (IKA, Staufen, Germany). The moisture content of ground products and semolina was determined according to the AACC 44-15.02 method [39], and their total protein content was determined according to the AACC 46-12.01 method [40] with 5.7 as nitrogen-to-protein conversion factor.

### 2.5. Protein Extraction Procedures

Proteins were extracted from lab-milled and stone-milled grains according to Morel et al. [41] with some modifications. The sample (160 mg) was suspended in 20 mL of 0.1 M sodium phosphate buffer (pH 6.9) containing 1% (*w/v*) sodium dodecyl sulfate (SDS) and shaken for 80 min at 60 °C. After centrifugation (37,000× *g*, 30 min, 20 °C), the supernatant was stored (−20 °C) until the Size Exclusion-High Performance Liquid Chromatography (SE-HPLC) analysis and the pellet was re-suspended in 5 mL of SDS-phosphate buffer before being sonicated for 3 min at 7.5 watts and centrifuged under the same conditions. The new supernatant was kept until analysis. For heat-treated products (dry and OCT-cooked pasta), the first extraction step was the same as before. However, the pellet was re-suspended in sodium phosphate-SDS buffer containing 20 mM of dithioerythritol (DTE) and the content of the tube purged with argon. The suspension was then stirred for 1 h at 60 °C before being sonicated for 3 min at 7.5 watts and then centrifuged. The supernatant was kept until the SE-HPLC analysis.

### 2.6. Measurement of Protein Size Distribution by SE-HPLC

The proteins obtained after the different extraction steps were separated by SE-HPLC using a TSKgel G4000 SWXL column (7.8 mm i.d. × 30 cm, TOSO BIOSCIENCE GmbH, Griesheim, Germany) according to Dachkevitch and Autran [42] on an Alliance system (Waters, Saint Quentin en Yvelines, France). Proteins were eluted at room temperature with 0.1 M sodium phosphate buffer (pH 6.9) containing 0.1% (*w*/*v*) SDS at a flow rate of 0.7 mL. min^−1^, and the absorbance was recorded at 214 nm. The first chromatogram corresponds to the SDS-soluble proteins and can be divided into five fractions, from F1 to F5, according to Morel et al. [41]. The F1 and F2 fractions include the large and small glutenin aggregates, respectively; F3 contains ω-gliadins and high-molecular-weight albumins such as β-amylase; F4 includes α-, β- and γ-gliadins; and F5 contains albumins–globulins. The total area under the second chromatogram of the non-thermally treated products was calculated and named Fi, for insoluble fraction. The total protein content of the samples was estimated from the sum of the total areas of the two chromatograms corresponding to the same sample. Fractions F1 to F5 and Fi were expressed as a percentage of this sum. The proportion of unextractable polymeric proteins (UPP%) was calculated as follows: UPP% = (Fi × 100)/(F1 + F2 + Fi). In the case of heat-treated products, the first chromatogram was not cut into 5 fractions; only the total area was retained to obtain SDS-soluble proteins expressed in % of the total proteins. The proportion of DTE-soluble proteins (i.e., after DTE reduction and sonication) was calculated from the total area of the second chromatogram and also expressed in % of the total proteins. The non-extracted protein fraction was calculated by subtracting the sum of the SDS-soluble and DTE-soluble protein contents from the Kjeldahl total protein content. When the fraction of non-extracted protein was negative, due to high recovery, this value was forced to 0, and the sum of the SDS soluble and DTE extracted protein fractions was corrected to reach 100%. The gliadin to glutenin ratio was calculated as follow: Glia/glu = (F4/(F1 + F2 + Fi)).

### 2.7. In Vitro Protein Digestibility

The digestibility of OCT cooked pasta samples was evaluated by measuring the rate of proteolysis in vitro using the Protein Digestibility Assay kit (Neogen, Auchincruive, UK), with a few modifications to the standard procedure. For each pasta sample, proteolysis was carried out on two 250 mg samples, according to manufacturer specifications and trypsin/chymotrypsin digestion was conducted for 3 h. Digestion was stopped by immersing the tubes in boiling water. After cooling, the tubes were centrifuged at 4696× *g* for 15 min at 15 °C. After centrifugation, the supernatants were set aside and the pellets frozen. Protein digestibility was then estimated by determining the amount of nitrogen remaining in the pellets from two extractions of the same sample using the Kjeldahl method. The extent of proteolysis after 1 h of peptic digestion followed by 3 h of tryptin/chymotryptin digestion was expressed as the percentage of the initial protein content of the sample that remained after digestion.

### 2.8. Statistical Processing

ANOVA, Multivariate analyzes of variance (MANOVA) and Pillai trace tests with a 95% confidence interval were carried out in order to assess the significance of the impacts of the factors studied on all the variables measured, and rank the influence of these factors. The significance of the differences between modalities was tested by SNK (Student–Newman–Keuls) tests within a 95% and 99% confidence interval.

Correlation analyses allowed performing existing links between the different variables. Both SAS Software (SAS 9.3 TS-Version windows 6.2.9200. SAS Institute Inc. SAS and all other SAS Institute Inc., Cary, NC, USA) and R packages were used to perform these analyses: readxl, xlsx, FactoMineR, Ggally, agricolae, tidyverse and rlist R Studio (version 4.0.2).

## 3. Results

### 3.1. Whole Grain Quality Criteria

Three main criteria—thousand-kernel weight (TKW), specific weight (SW) and protein content (PROT)—usually considered in the case of durum wheat commercial contracts have been estimated on whole grains by NIRS. These estimates have been made on the whole grain samples of 4, 18 or 20 varieties according to the year. Five environments were considered: 3 years of harvest (2016, 2017, 2019) and, for some years, 2 farming systems (ZI and LI) or 2 different locations (Mauguio and Purpan).

A classic ANOVA comparing the impact of the varieties and the impact of the environments on these data made it possible to define the contribution of each of the factors to the variability of the criteria (Table 3).

As expected, TKW and SW were mainly heritable criteria (the variety effect explained 67% and 68% of the variability, respectively) while the environment played a less role (20 to 29% of the total variability), whereas the protein content was equally dependent on the variety (42%) and the environment (48%).

#### 3.1.1. Thousand-Kernel Weight

Across all years, the TKW of FAR varieties was significantly higher than that of IND and DIV varieties (Table 3). For instance, in H2016, the TKW varied from 37.9 g (cv. RG218) to 51.9 g (cv. RG1110). The varieties used by the farmers showed, on average, higher TKW (49.6 g) than those recommended by the industrial sector (40.8 g) and, the others underutilized genetic resources (42.6 g).

Only four varieties were tested in 2019 (H2019) in two different locations (Mauguio and Purpan); their TKW varied, on average, from 47.5 g (cv. Anvergur) to 60.2 g (cv. LA1823). The TKW was higher in Purpan (mean = 56.6 g) than in Mauguio (mean = 51.4 g). When selecting the data for these 4 varieties from the others years, we noted also a higher and significant TKW for the two varieties used by the farmers (“FAR”) than for the two used by the industrial sector (“IND”), regardless of the year and cropping conditions.

#### 3.1.2. Specific Weight

The specific weight is an essential data factor to be considered in case of commercial contracts. It varied on average, from 80.7 to 86.7 kg.hL^−1^ (Table 3). Only in 2017 was the usual threshold of 76 kg.hL^−1^ not reached for one variety (cv. Miradoux).

Whatever the year of production and the cropping system or location, the specific weight of the varieties used by the farmers (overall mean FAR = 85.2 kg.hL^−1^) was always greater than that of the varieties recommended and used by the industrial sector (overall mean IND = 82.2 kg.hL^−1^). However, no significant difference was found with varieties classified in “DIV” (overall mean DIV = 82.6 kg.hL^−1^).

The low input cropping system implemented in 2017 led to a higher specific weight (83.5 kg.hL^−1^) than the zero-input cropping system (80.7 kg.hL^−1^).

In H2019, for the four studied varieties, the average specific weight varied between 84.4 kg.hL^−1^ (cv. Anvergur) and 87.0 kg.hL^−1^ (cv. LA1823). The specific weight was, on average, higher at Mauguio (86.7 kg.hL^−1^) compared to Purpan (84.7 kg.hL^−1^).

#### 3.1.3. Protein Content

The protein content of the grains harvested in 2016 varied from 7.5% db (cv. Karur) to 10.6% db (cv. RG218). The varieties used by farmers and artisans (FAR) contained significantly more proteins (mean = 9.7% db) than the varieties recommended by the industrial sector (IND) (8.6% db) (Table 3).

On the 2017 harvest, protein levels were higher (14.1% db for FAR and 12.7% db for IND) than in H2016 despite the higher grain yield. There was no noticeable effect of cropping system (ZI or LI) on protein content.

Concerning the grains harvested in 2019, at the Mauguio site, the protein level was lower than in H2017 (11.3% db vs. 13.6% db, respectively), when the same four varieties were considered. On the Purpan site, the protein content was slightly lower (10.2% db) than at Mauguio.

Similar observations were noted in each environment: the varieties used by the farmers presented a higher protein content (general mean FAR = 12.5% db) than those used by the industrial sector (general mean IND = 11.0% db). This has to be linked with the level of the grain yield, which was significantly lower for FAR (2.4 t.ha^−1^) than for IND (3.2 t.ha^−1^) (Figure 1).

### 3.2. Whole Grain Protein Composition

Samples of harvested grains (H2016, H2017, H2019) were ground in a laboratory mill, and the protein analysis was performed using the SE-HPLC method. Six protein fractions (F1 to F5 and Fi) were obtained after SE-HPLC separation (Appendix A).

The analysis of variance of these fractions allowed us to identify the main factors responsible of the variability of each of them (Table 4).

Except for fraction F1, all the others fractions varied mainly according to the year. Then, the factor “variety” also contributed in large part to the variability of all the fractions but to varying degrees.

When considering only the year 2017, the main part of the variability of the criteria was due to the varieties (Table 4).

Variety largely explained the variance of the protein fractions (Table 4). The Cropping system effect and the Variety × Cropping system interaction help to explain F2 and Fi fractions. The ANOVA on fractions F1, F3 and F5 showed higher statistical residuals (around 20% of the explained variance, compared to less than 10% for the other fractions), meaning that others unexplained factors may have an impact on these fractions.

As expected, the major part of the grain proteins was α-, β- and γ-gliadins (F4 fraction), followed by small glutenin aggregates proteins (F2) and albumins and globulins (F5) (Figure 2).

#### 3.2.1. Year Effect

Except for fraction F1, significant differences appeared for the others fractions between the 2 years: H2016 (Figure 2, left) and H2017 (Figure 2, right). A significant increase was noted in the insoluble fraction (Fi) (from 6% in 2016 to 11% in 2017) and in F4 (from 29% to 36%). This compensated for a decrease in F2 (from 23% to 18%) and F5 (from 23% to 18%).

It seems that the higher protein content measured in 2017 (+2.6%) was due to gliadins and unextractable proteins.

To assess more deeply the year effect, Table 5 includes only the data concerning the zero-input cropping system (present the 3 years) and the data concerning the four varieties common to the 3 years.

The year effect was very high for all the parameters and higher than the variety effect except for the F1 fraction.

#### 3.2.2. Cropping System Effect

When comparing the two cropping systems (zero input, ZI; low input, LI) implemented only in 2017 in Mauguio (Table 6), the Fi and F5 fractions were significantly higher in zero input (11.1% and 18.3%) than in low input (9.6% and 17.8%), respectively, while the percentages of large and small glutenin aggregates (F1 and F2) were significantly lower in zero input (6.9% and 18.5%) than in low input (7.3% and 19.3%). The others fractions (F3 and F4) were not significantly different.

The contribution of the cropping system effect to the protein composition is mentioned Table 6. The cropping system had a significant effect on total protein content, F2, Fi and UPP and to a lesser extent on F1 and F5. The fractions F3 and F4 and the ratio gliadin to glutenin (Glia/Glu) were not impacted by the considered cropping system.

#### 3.2.3. Location Effect

Only the data of H2019 are available to assess the impact of the location (Mauguio vs. Purpan) on the protein quality parameters. Data are presented Table 6. The location had an important impact on protein content, then on F1, F3, F4 and F5 fractions. No effect was observed on F2, Fi and UPP. In these two locations, durum wheat was grown under zero-input condition.

#### 3.2.4. Variety Effect

Concerning the varieties, the major coefficients of variation were obtained for Fi and F1 fractions (Figure 2). When zooming on each of these protein fractions measured in 2016 (Figure 3), the percentage of Fi varied from 3.3% (cv. Oued Zenati) to 8.5% (cv. Claudio). Compared to the average of the 19 varieties, with a highest Fi content Claudio had, conversely, a lower proportion of soluble glutenin aggregates (F1 and F2, 6.2 and 20.6%, respectively) and of ω-gliadins and high-molecular-weight albumins (F3). Other varieties, such as cv. Oued Zenati, were characterized by a higher proportion of F1 and F2 (6.8 and 23.3%) and a lower proportion of Fi (3.3%). The Vivit variety was distinguished by the lowest proportion of F4 (27.0%) and a significant amount of albumins and globulins (F5) (24.0%).

#### 3.2.5. Comparison of Farmer Varieties (FAR) with Industry Recommended Varieties (IND)

The comparison of the means of protein fractions measured on the varieties used by the farmers (FAR) and those recommended by the industry (IND) showed significant differences, except for F1 (Table 7).

The higher protein level noted on the FARs seemed essentially due to an increase in the F2 and F4 fractions, whereas the fraction of insoluble proteins Fi was slightly higher for the IND varieties (Table 7).

The same trend, i.e., significant higher F4 fractions and lower F5 and Fi fractions in FAR varieties compared to IND varieties, was systematically observed regardless of the environment (Figure 4).

### 3.3. Protein Composition of Stone-Milled Grains (Semolina)

Measurements on semolina were carried out for four varieties harvested in 2017: Anvergur, Claudio (IND) and Bidi17, LA1823 (FAR). This semolina was produced at Lycée Agricole d’Auzeville on a Tyrol stone-mill (SM 2).

The total protein content of stone-milled grains samples assessed by the Kjeldahl method was similar to those measured on whole grain at harvest by NIRS (except for cv. Anvergur and Claudio, which have a slightly higher—2 points—NIRS protein content) (Appendix A).

The protein composition of semolina obtained from stone-milled grains is presented in Appendix A. The results are very similar to those measured on grains ground with a laboratory (lab-milled) (Figure 5).

The F1, F2 and F4 fractions are very similar between the two measurements (lab-milled grains and stone-milled grains), they are correlated at 81%, 92% and 84%, respectively. However, the F3 fraction appears less well correlated (6%), and the correlations of the F5 and Fi fractions are lower (59% and 49%). When considering the whole data set, the correlation is high (*r*^2^ = 0.99).

The contribution of each factor to the variability of each parameter analyzed in stone-milled semolina is mentioned in Table 8.

Compared with the percentage obtained on lab-milled grains (Table 3), we noted a similar impact of each factor for the protein content, a greater impact here of the factor “variety” except for F3, F4 and F5, a lesser importance of the environment, except for F5, and a greater impact of the interaction Variety × Cropping system, notably for F3, F4 and F5.

Regarding the type of variety (Appendix A), for all the fractions, except F4, significant differences exist between varieties used by the farmers (FAR) and those recommended by the industry (IND). The latter have fractions F3, F5 and Fi higher than FAR varieties, and less medium and small glutenin aggregates (F1, F2).

### 3.4. Protein Composition of Dry Pasta

The semolina (stone-milled grains) of the 4 varieties were used to make pasta, according to two types of process: “Industrial” and “Artisanal 2”. The “Industrial” process differed from the “Artisanal 2” by 2 points: the extrusion was under vacuum, temperature controlled, and the drying of the pasta was performed at very high temperature (90 °C instead of 37 °C for the artisanal process) (Table 2). The protein composition of dry pasta was analyzed by SE-HPLC (Appendix A). Because the extrusion and drying processes induced a decrease in solubility of some proteins, DTE was used to recover them, and the Fi fraction was renamed “DTE-soluble” fraction. The contribution of the varieties, cropping systems and process to the variability of the protein fractions in dry pasta is mentioned Table 9.

For fractions F1 to F5, the process alone (Artisanal 2 vs. Industrial) explained between 88 and 98% of the variability. For fraction F5 (albumin and globulin), the interaction Variety × cropping system was significant. The unextractable proteins were depending on the process, on the variety and on their interactions.

Even if the varieties explained very few of the variability of the protein fractions, when grouping on one hand the two varieties used by the farmers (FAR) (cv. LA1823, cv. BIDI17) and on the other hand, the two others recommended by the industry (IND) (cv. Anvergur and cv. Claudio), we noted significant differences mainly for the fractions of proteins soluble only in the DTE (34.6% for FAR and 37% for IND), and for the unextractable proteins (4% for FAR and 1.5% for IND).

The impact of the process was major on the proportion of the protein fractions in the dry pasta. Whatever the cropping system (zero input, ZI, or low input, LI), the “Industrial” process led to a large increase in the DTE-soluble fraction (DTE-soluble represented 53.9% of the total protein in “Industrial” process vs. 19.3% in the “Artisanal 2” process) and to a decrease mostly in the F2 (26% of reduction) and F4 fractions (Figure 6).

Under the zero input condition, the two varieties Bidi17 and LA1823, used mainly by the artisanal sector, showed a greater difference than the two other varieties (cv. Anvergur and cv. Claudio) for the F2 and DTE-soluble fractions. In low-input (LI) conditions, the difference was not significant for DTE-soluble fraction (Appendix A).

### 3.5. Comparison between Lab-Milled Grains, Stone-Milled Grains (Semolina) and Dry Pasta (Artisanal 2 and Industrial Process) under the Cropping System Zero Input (ZI)

The comparison between the distribution of the fractions measured in the semolina (stone-milled grains) and in the dry pasta produced according to the two processing modes (Artisanal 2 and Industrial) highlighted the impact of the mixing, extrusion and drying stages (Figure 7).

The main differences were the important decrease in the F1 fraction, significantly reduced by the IND pasta process and the large increase in the DTE-soluble fraction. The other fractions varied little between artisanal dry pasta and semolina.

When adding lab-milled grains to the comparison, only F1, F2 and F4 fractions were well correlated with those of stone-milled grains (Table 10). Surprisingly, F3, F5 and Fi (or DTE-soluble) were not correlated. No correlation was found between lab-milled grains and dry pasta fractions.

### 3.6. Protein Composition of Cooked Pasta

Protein fractions were also analyzed by SE-HPLC on two different sets of pasta cooked at OCT.

The first set of pasta was obtained from ten varieties of durum wheat chosen among the 20 harvested at the Mauguio site, in 2016 (Table 1), milled and processed by a farmer (Table 2 = Artisanal 1).

The second set was composed of pasta produced with four varieties of durum wheat (Anvergur, Claudio, Bidi 17, LA1823), processed by three methods: “Industrial”, “Artisanal 1” or “Artisanal 2” (Table 2). The results of the dry pasta of this second set were presented in the previous section.

Because cooked pastas are heat-treated products, the first chromatogram obtained from each sample was not cut into five fractions; only the total area was retained to obtain SDS-soluble proteins expressed in % of the total proteins. The proportion of DTE-soluble proteins (i.e., after DTE reduction and sonication) was obtained from the total area of the second chromatogram and also expressed in % of the total proteins. The difference between the protein contents calculated from the total areas and the protein content measured by Kjeldahl allowed us to deduce the unextractable part of proteins (unextractable). It should be noticed that, with varieties processed according to the “Artisanal 1” process, no unextractable fraction was recovered (Appendix A).

#### 3.6.1. First Set: Proteins Composition of the Cooked Artisanal Pasta Made from 10 Varieties of Durum Wheat

The 10 samples of cooked pasta, made from the grains of the varieties harvested in 2016, showed significant differences in the fraction of soluble proteins in the SDS: from 34.3% (cv. Karur) to 48% (cv. Pescadou) and conversely for DTE-soluble proteins (Figure 8). There was no correlation between protein content and the others parameters (Appendix A).

Although there was, in this set, an imbalance between the number of varieties recommended by the industry (eight varieties) and those used by the farmers (two varieties), we did not observe differences between these two types on the protein fractions. However, the difference regarding the protein content remained (Appendix A).

#### 3.6.2. Second Set: Proteins Composition of the Cooked Pasta Coming from Four Varieties of Durum Wheat Processed with Two Methods: “Industrial” and “Artisanal”, and In Vitro Proteolysis Data

Four varieties, among which were cv. Anvergur and cv. Claudio, recommended by the industry, and cv. LA1823 and cv. Bidi17, used by the farmers, were harvested in 2017, in the Mauguio site, then stone-milled and then processed according to two types of methods: “Industrial” and “Artisanal 2”. The “Industrial” process differed from the “Artisanal 2” process in two ways: the extrusion that was under vacuum and temperature controlled, and the drying of pasta that was made at a very high temperature (90 °C instead of 37 °C or ambient temperature for the artisanal process). In this results section, pasta were produced according to two artisanal processes: “Artisanal 1” on wheat grains harvested in 2016 (see results above), and “Artisanal 2” on wheat grains harvested in 2017 (Table 2).

In order to compare the effect of process, we considered, in Table 11, only the data coming from “Artisanal 2” and “Industrial” processes.

The ANOVA highlighted the importance of the factor “Process” on proteins soluble in SDS and DTE (Table 11) and the factor “variety” had an impact on the amount of unextractable proteins and total protein content measured on cooked pasta.

The four varieties exhibited a total protein content ranging from 12 (cv. Anvergur) to 15% db (cv. Bidi 17). The quantity of proteins solubilized with DTE ranged in average from 61.5 (cv. Claudio) to 68.7% (cv. LA1823), and those that were unextractable from 6.9% to 14.8% (Appendix A).

The cropping system, zero input or low input, did not have an impact on the considered parameters except on total protein content.

The protein content was significantly and negatively correlated with DTE-soluble protein content (*r*^2^ = −0.54) at 0.0001% and with in vitro-digested proteins (*r*^2^ = −0.63).

When also considering the data coming from the “Artisanal 1” process, we confirmed the importance of the process (Figure 9) on the protein solubility.

Both “Artisanal” processes led to pasta containing the most easily soluble proteins (33.3% of SDS-soluble, against 19.9% for industrial pasta) (Figure 9 and Appendix A). Consequently, “Industrial” pasta presented a high level of highly polymerized proteins, soluble only with DTE. No significant difference was observed for the unextractable proteins.

An in vitro test was carried out to measure the protein digestibility of OCT cooked pasta. It was performed with gastric and intestinal enzymes (pepsin, chymotrypsin and trypsin). The amount of digested proteins was measured at the end of the digestion step (= 100 − % of proteins remaining in the solid portion of the sample). The quantity of digested proteins was significantly higher in artisanal pasta (36.2% on average) than in industrial ones (28.6%) (Appendix A).

## 4. Discussion

Survey data and one-on-one interviews [18] highlighted that some consumers, claiming to suffer from gastric disorders, consume pasta again when it comes from farmers or artisanal short chains. Such products are most often elaborated from local or old varieties, grown organically. The grains are then processed on the farm, ground with stone-mills and used to make pasta, produced under “slow” conditions, avoiding, for instance, drying at high temperatures, as is performed in industry.

Starting from these observations, the objectives of this study were to evaluate the importance of various factors: variety, cropping system, process method on a major component of cereals, and proteins, in particular, those of gluten, identified in the literature as possible NCGS/NCWS (non-coeliac gluten sensitivity/non-coeliac wheat sensitivity) triggers. To date, no study has made it possible to link with certainty any constituent of wheat to NCGS/NCWS (i.e., gluten, ATIs, FODMAPs, wheat germ agglutinin) [43,44,45,46]. This study was not intended to establish this link.

The differences between artisanal and industrial type products were therefore approached from the angle of the protein composition of the grains and semolina, the degree of aggregation/reticulation of the proteins in the pasta following the process and their susceptibility to in-vitro proteolysis.

### 4.1. Evolution of Quality Parameters from Grain to Pasta

Wheat proteins undergo structural changes during the pasta making process due to changes in hydration during mixing and drying, the input of mechanical energy during mixing and extrusion, and the intensity of heat treatment during drying. During the mixing phase, the semolina particles are hydrated, and the mobility of the compounds, including that of the proteins, is favored. It is only then, during extrusion, under the effect of mechanical stress, that glutenins and gliadins interact and form intra- and inter-molecular disulfide bonds that constitute the network of gluten [47,48].

Regarding the 10 varieties for which we have data on lab-milled grains and cooked pasta, the protein content of ground grains is, as expected, predictive of the protein content of cooked pasta (correlation coefficient *r*^2^ = 0.88). The F5 fraction measured on ground grains also seems to be a fairly good predictor of the amount of soluble protein in SDS (*r*^2^ = 0.63).

When considering the four varieties (Anvergur, Claudio, LA1823, Bidi17) for which we have data on ground grains, semolina, dry pasta and cooked pasta, we observe that the protein content of cooked pasta is very correlated to the total protein content present in grain (*r*^2^ = 0.92) or in dry pasta (*r*^2^ = 0.87). The total protein content of ground grains is also significantly and negatively correlated with the F3 (*r*^2^ = −0.78) and F5 (*r*^2^ = −0.86) fractions, and it is positively correlated with fraction F4 (*r*^2^ = 0.85) analyzed on ground grains.

Between the lab-milled grains and the stone-milled grains (semolina), the F1, F2 and F4 fractions are strongly correlated (*r*^2^ = +0.89 to 0.93).

Due to thermal treatment, the fractions analyzed on dry pasta are not correlated with those measured on lab-milled grains.

The amount of SDS-soluble protein of cooked pasta is strongly and positively correlated (*r*^2^ = +0.89 to 0.95) with the F1 to F4 fractions measured on dry pasta. Thus, it is also strongly correlated with the amount of SDS-soluble protein (sum of fractions F1 to F5) measured on dry pasta (+0.93) and inversely correlated with the DTE-soluble protein content of dry pasta (−0.92). The other parameters measured on cooked pasta, such as DTE-soluble proteins, unextractable proteins, and protein digested in vitro, show no significant correlation with other parameters.

In the artisanal process, with drying at a low temperature (37 °C), a slight decrease in the solubility of proteins is observed on the dry pasta, in comparison with the semolina, in particular on the F1 fraction, concomitant with an increase in the insoluble fraction. This loss of protein solubility between semolina and dry pasta reflects a change in protein behavior despite a “slow” processing mode. The quantity of SDS-soluble proteins thus goes from 89.2% in the semolina to 80.1% in the dry artisanal pasta. This decrease is significant and consistent with the conclusion of Aktan & Khan [30] and De Zorzi et al. [49], who showed that drying induces a decrease in protein solubility. However, it is not in agreement with the work of Wagner et al. [50], who showed no significant difference in SDS-soluble proteins between semolina and fresh pasta (produced with the same equipment and under the same extrusion conditions as our industrial-type pasta). This result is to be compared with that of Martin et al. [51], who showed an insolubilization of glutenins and to a lesser extent of gliadins during resting phases experienced at 20, 30 or 40 °C and applied before drying. These authors indicate that there are no further differences in protein solubility between the pasta that have undergone a resting period and those that have not, as soon as they are all dried at a high temperature.

### 4.2. Differences between Varieties Used by Farmers and Those Recommended by Industrialists

The French pasta companies annually update a list of around 10 varieties chosen from among the 568 varieties of durum wheat currently registered in the European catalogue. In 2010, they created a network of field trials called “Observatory” to “assess the agronomic aptitudes impacting the physical traits and technological qualities of the main cultivated varieties and those potentially in the making. Pasta and semolina manufacturers thus intend to transcribe precisely, in particular with breeders, the needs of processing industries to all actors in the durum wheat sector”. Among the technological criteria, they are mainly interested in semolina yield, semolina and pasta aspect, and cooking quality.

Farmers who mill their own harvest and turn it into pasta, use a small number of varieties of durum wheat. They are looking for locally adapted varieties or heritage varieties that give a particular flavor to the pasta. They often considered their varieties as “old” even the cv. LA1823, which comes from a participatory plant breeding project [52] and was released in 2006!

It was therefore difficult to obtain a sample of varieties representative of the diversity cultivated for each sector and which is balanced for statistical analyses.

The varieties recommended by the French industry of semolina and pasta and the varieties used by peasant pasta makers differ significantly on several quality criteria measured on their grains such as TKW, SW, and protein content. Concerning TKW and SW, our data show that they are higher for varieties used by smallholder processors compared to the recent varieties used by industry. This could be explained by a shorter filling period for modern varieties. Thus, Nazco et al. [53] reported a 12% reduction in the duration of filling of modern varieties compared to landraces, which would explain their lower weight.

If we consider the protein content, the results of the first set of data, coming from the analysis of the grains of 20 varieties and the cooked pasta made from 10 varieties chosen among the 20 (“Artisanal 1”), show a higher protein content for the varieties used by peasant pasta makers.

The yield/protein dilution curve (Figure 1) confirms that this protein level is linked to the lower yields obtained for these varieties. This can be explained by the role of breeding which reduced grain protein concentration due to improved yields, but without impacting pasta cooking quality [54,55].

We confirmed these results thanks to the second set of data, comprising four varieties for which we have data on each of the following products: lab-milled grains, stone-milled grains (semolina), dry pasta and cooked pasta. Whatever the product, its protein content is always significantly higher for the varieties used by peasant pasta makers.

The proportion of the different protein classes is different between varieties recommended by the industry and the farmers. We observe that the increase in the protein content in the varieties used by farmers is expressed mainly by an increase in the F1, F2 and F4 fractions, to the detriment of the F5 fraction. Concerning the F4 fraction (α-, β- and γ-gliadins), the superiority measured in ground grains of the varieties processed by the farmers is consistent with the literature synthesis of Suter and Békés [56], stating that “Because of the trend of decreases in overall protein content […] older varieties are higher in gliadin content”. It must be added that De Santis et al. [12] or Pronin et al. [13] noted no significant differences in the expression of α-, β- and γ-gliadins (F4) between modern and old genotypes but mentioned a significant decrease in ω-gliadins (F3) in the modern genotypes.

Breeding programs have aimed to improve technological quality by selecting certain glutenin alleles and increasing glutenins quantity [12,13,57,58]. The new varieties are characterized by a better expression of LMW-GS, allowing us to obtain a lower Glia/Glu ratio and therefore to increase the strength of the gluten [12,59]. According to Mefleh et al. [60], the genotypic variation in grain protein percentage among old varieties was more strongly associated with glutenin than with gliadin content. We do not note any difference concerning the ratio Glia/Glu between varieties used by farmers and those used by the industry probably due to too few varieties studied in both categories.

In another article, De Santis et al. [14] reported a content of unextractable polymeric protein (UPP) twice as high for the modern variety Saragolla as for the old variety Cappelli. Our data also show an increase in the proportion of UPP in the grains and semolina of the varieties recommended by industry, a proportion related, according to Sissons et al. [25], to the strength of gluten. This is correlated with the level of DTE-soluble proteins in dry pasta, which is also higher in the varieties recommended by the industry. However, after cooking, we do not see anymore any differences for this parameter.

The four varieties tested in 2017 do not differ in the Glu/Glia ratio between modern and old genotypes. On the contrary, in 2016, the Glu/Glia ratio was indeed higher for the modern genotypes on milled grain.

Several studies comparing modern wheats and ancient wheats have attempted to link the harmful effects of gluten to the year of creation and the change in protein composition. Some authors, such as Branlard and coworkers [61,62], have suggested that modern breeding would have led to varieties with more glutenin polymers or larger polymers, stronger glutens and therefore potentially more difficult to digest.

It appears that old varieties are not all less toxic than recent varieties [15,16,17]. However, the study by Ianiro et al. [18] shows that patients with NCGS showed fewer gastrointestinal and extra-intestinal symptoms after ingesting pasta made with an “old” variety of durum wheat (Senatore Cappelli, date of release = 1923) compared to ingesting pasta from a mixture of recent varieties. This is consistent with the behavior observed among customers of the farmer pasta makers involved in our study.

It would seem that the division of IND vs. FAR, in our study, is of little relevance to explain the great variability observed between the varieties on the quality of the proteins and their proteolysis in vitro. Indeed, concerning the unextractable proteins in cooked pasta, the cv. Claudio (IND) and Bidi 17 (FAR) have the highest percentage, whereas cv. Anvergur (IND) and LA1823 (FAR) have the lowest. Moreover, the results of protein solubility, obtained on pasta produced using a single process, do not make it possible to oppose the varieties used by farmers/artisans (FAR) and those recommended by the industry (IND).

### 4.3. Genotypic Variation for Quality Parameters

Our results on protein solubility (SDS- and DTE-soluble proteins) of cooked pasta produced by the same farmer (same milling and processing conditions, set 1) show a different aggregation state depending on the variety.

Indeed, there are more SDS-soluble proteins from cooked pasta made with cv. Bidi17, Pescadou and Miradoux. This difference can, perhaps, be explained by the glutenin composition.

Most of the varieties with high SDS-soluble protein content have the HMW-GS type 7 + 8 encoded at the Glu-B1 locus (Pescadou, Miradoux, Claudio), while those with a lower proportion of SDS-soluble proteins have the 6 + 8 type (Joyau, Karur). In between are varieties with the HMW-GS 20x + 20y type (Bidi17, Qualidou, LA1823) or 13 + 16 type (Fabulis, Anvergur). This could allow us to link a weak and less cohesive gluten to more SDS-soluble proteins. However, this would only be a partial explanation, as in durum wheat, the work of Carrillo and colleagues has shown that it is most often the HMW-GS 20x + 20y type varieties that have a weaker gluten, and conversely, those of 7 + 8 or 6 + 8 types have medium to high gluten (see [63,64]). Moreover, HMW-GS composition alone is not sufficient to predict the gluten strength, as it is accepted that group B LMW-GS remain the key factor determining factor of gluten quality in durum wheat.

Determination of the allelic composition of group B LMW-GS could help to better explain protein solubility in relation to gluten strength.

### 4.4. Effect of Environment on Protein Content and Composition

The comparison of the 3 years of culture shows that the biotic and abiotic environmental factors play a determining role in the protein content and the composition of the grains of durum wheat, compared to the variety.

The environmental variables considered in our study were: year, location and cropping system. For harvested grain, protein content was highest in 2019, followed by 2017 and 2016. The interaction between year and variety was significant for this parameter.

Protein composition measured on lab-milled grains was also greatly affected by the year, with the exception of the F1 fraction (large glutenin aggregates), which is mainly variety dependent. The ratio Glia/Glu and the F4 fraction are also affected by the interaction year x var. These results are in line with those of Zivancev et al. [65], who observed that the year of production has no significant effect on the percentage ranges for glutenins. However, they had a considerable impact on the percentage ranges for gliadins.

Climatic conditions (heat stress and drought) and cropping system are known to affect protein synthesis leading to variation in strength of gluten and affecting the final quality of semolina and flour [66]. Thus, Labuschagne et al. [66] showed that in response to stress conditions (high temperature and drought) SDS-extractable HMW-GS and gliadins increase, while SDS-unextractable HMW-GS decrease. DuPont et al. [67] show that under a high temperature regimen (37/28 °C) and with the addition of NPK, the total protein content had increased, mainly due to an increase in ω-gliadin and HMW-GS.

When comparing the two cropping systems (zero input, ZI; low input, LI) implemented only in 2017, significant differences were noted on total protein content, F2, Fi and UPP, and to a lesser extent on F1 and F5 measured on lab-milled grains. The lower total protein content obtained with ZI results in a lower level of glutenins.

These cropping systems were little contrasted: in both cases, the previous crop was a legume, the only difference was a nitrogen supply of 150 u and the use of a herbicide for the LI. However, these results are partially in line with those of Park et al. [68], who concluded that N fertilizer increased flour protein, relative concentration of glutenin and relative amount of high-molecular-weight glutenin subunits (HMW-GS/LMW-GS). Trials on more contrasting cropping systems over several years would make it possible to more accurately assess the impact of the cropping system.

We only have measurements on whole and ground grains to assess the effect of the location of grain production (Mauguio and Purpan in 2019). This factor has a significant effect on the total quantity of proteins, and to a lesser extent on some fractions.

Numerous authors [12,53,59] showed that the environmental factor was the main source of variation in protein content for modern varieties. Criteria such as rainfall, drought, sowing date and nitrogen fertilization can cause significant differences in grain protein content [69,70,71,72].

### 4.5. Impact of Transformation Process on Protein Profile of Pasta

After discussing the role of variety and the role of the environment, we are interested here in the impact of the type of processing on the protein status.

Among the studied factors (variety, year, location, cropping system, and processing), our results show that the processing is the main factor affecting the protein fractions and in particular the protein solubility.

The comparison of the types of processes implemented in this study, two different “Artisanal” types and one “Industrial” type, shows that the latter reduces the solubility of proteins in the SDS more strongly than the “Artisanal” types and that this difference remains, despite the cooking of the pasta. This confirms the results of Wagner et al. [50], indicating that the protein network formed during extrusion is largely insolubilized during drying and the greatest loss of protein solubility therefore occurs during the drying step. Our data, on dry pasta, show that the industrial process implemented, with the application of a very high temperature during drying, leads to a loss of protein solubility in the SDS. This data are in line with those obtained by Wagner et al. [50], Petitot et al. [73] or Bruneel et al. [74]. 

The heat treatment during cooking further reduces this solubility to 19.9%, close to the values of Wagner et al. [50], Petitot et al. [73] and Brunel et al. [74], which range from 13 to 18%. Thus, the aggregation of gluten proteins during the cooking of pasta reinforces the density of the protein network by the formation of an intermolecular covalent cross-linking inducing an additional loss of solubility of the proteins in the SDS. With low temperature drying, the amount of SDS-soluble proteins in cooked artisanal pasta is higher than in industrial pasta, despite the cooking step.

This difference in SDS solubility seems to be correlated with the degree of in vitro proteolysis. This rate is higher for artisanal pasta than for industrial pasta. These results agree with those of Petitot et al. [73], who show that the digestion of proteins is increased for pasta dried at a lower temperature compared to that of cooked pasta dried at a very high temperature.

## 5. Conclusions

The role of varieties, and in particular the difference between the so-called modern varieties and the so-called old varieties, is much debated regarding the quality of gluten. By comparing, in average, the varieties recommended by the industry (IND) and those used by farmers who process pasta (FAR), we observed that only the total amount of protein varies: the FAR being, on average, much richer in protein. However, the solubility of these proteins, their proteolysis in vitro by digestive enzymes and the fractions analyzed by SE-HPLC vary little between the two groups of varieties, while differences between varieties inside each group are greatly observable. The impact of the location of production and the tested cropping systems on protein quality was low. Yet, more contrasting modalities should be studied before confirming or invalidating this point. The pasta production process, approached here by artisanal-type production and industrial-type production, is, among those studied, the factor having the greatest impact on protein quality. Pasta produced by the artisanal method contains a higher SDS-soluble protein fraction and is more easily proteolyzed. Whether these criteria are indicative of what happens during a consumer’s digestion remains to be determined. It also remains to be assessed which key stages of the process have the greatest influence on protein quality.

## Figures and Tables

**Figure 1 foods-12-01093-f001:**
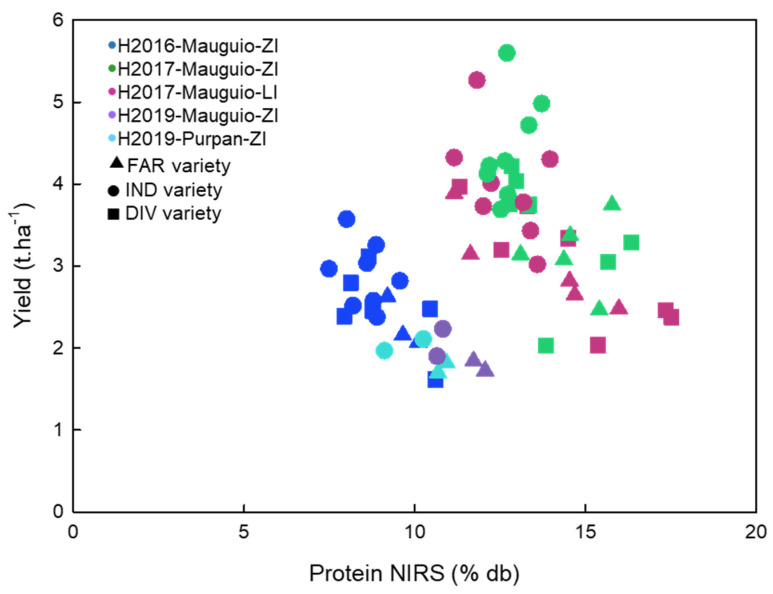
Yield (t.ha^−1^) vs. Protein content (% db NIRS) in the 5 environments. FAR: varieties used by the farmer’ pasta makers; IND: varieties recommended by the industry; DIV: underutilized genetic resources.

**Figure 2 foods-12-01093-f002:**
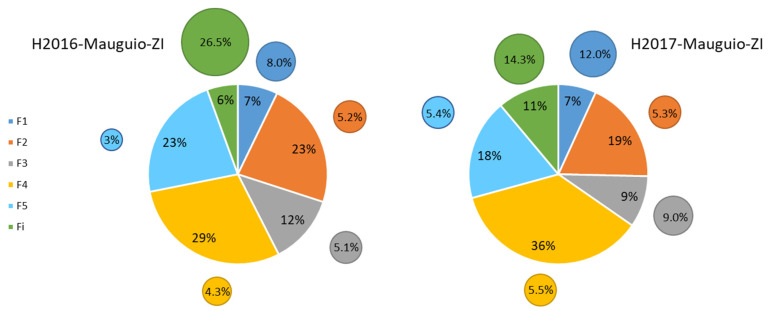
Comparison of 2 years (H2016, **left**; H2017, **right**), same cropping system (ZI) and same location (Mauguio). Proportion of the different protein fractions (F1 to Fi) in lab-milled grains. The size of the small outer circles is proportional to the varietal coefficient of variation of each fraction.

**Figure 3 foods-12-01093-f003:**
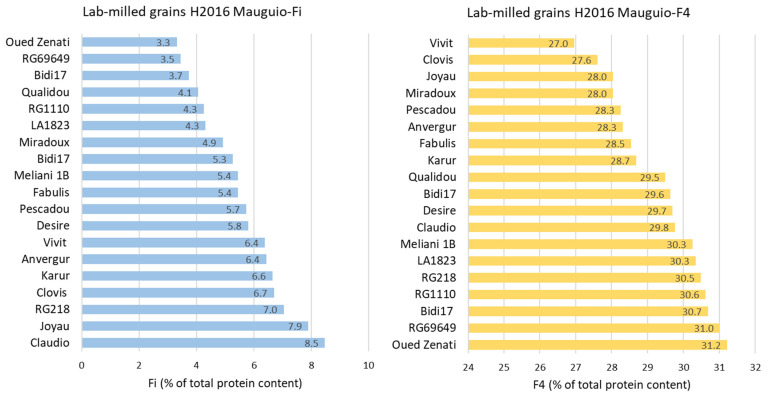
Proportion of Fi (**left**) and F4 (**right**) fractions measured on lab-milled grains of 20 varieties, cultivated under zero input and harvested in 2016 in Mauguio.

**Figure 4 foods-12-01093-f004:**
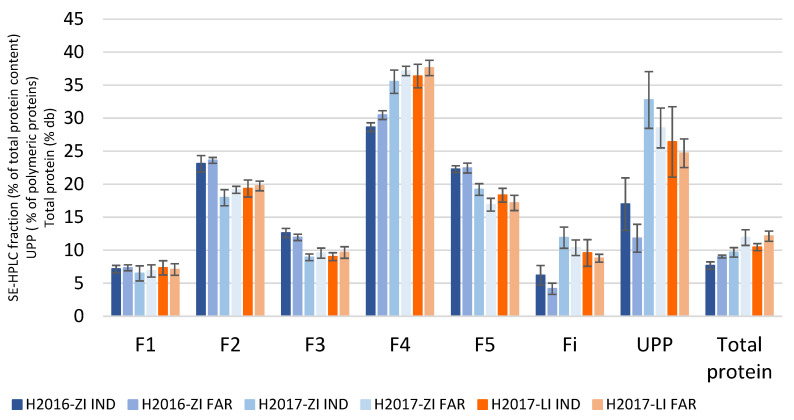
Comparison of the protein fractions measured on ground grains, cultivated in zero (ZI)- or low (LI)-input conditions and harvested in 2016 (H2016) or 2017 (H2017) in Mauguio. For each fraction, the means and error bars (SD) of the two groups of varieties are compared: IND = varieties recommended by the industry; FAR = varieties used by the farmers making pasta.

**Figure 5 foods-12-01093-f005:**
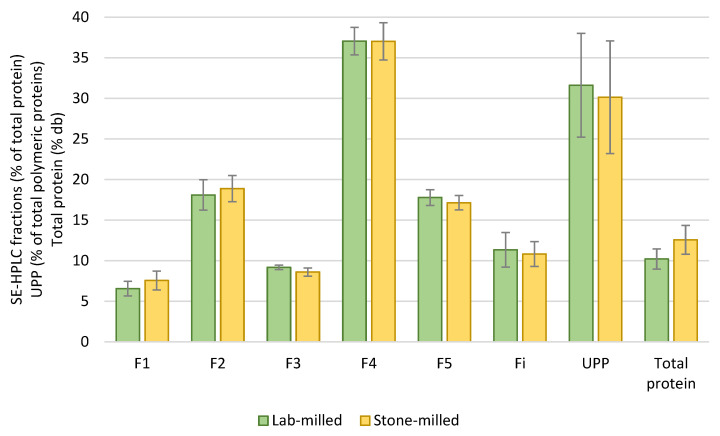
Comparison of the protein fractions measured on lab-milled grains and on stone-milled grains (or semolina). Means and error bars (SD) of 4 varieties: Anvergur, Claudio (IND) and Bidi17, LA1823 (FAR) harvested in 2017 (H2017) and grown under ZI conditions.

**Figure 6 foods-12-01093-f006:**
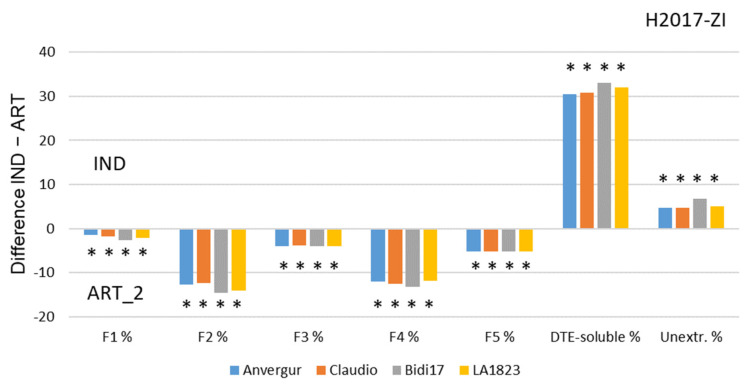
Differences between protein fractions (F1 to F5, DTE-soluble and unextractable) between dry pasta made according “Artisanal 2” and “Industrial” processes with stone-milled grains grown under ZI cropping system. When the bar is directed upwards, the effect is greater with the IND process; when it is directed downwards, the effect is greater with the ART_2 process. * Difference significant at 5%.

**Figure 7 foods-12-01093-f007:**
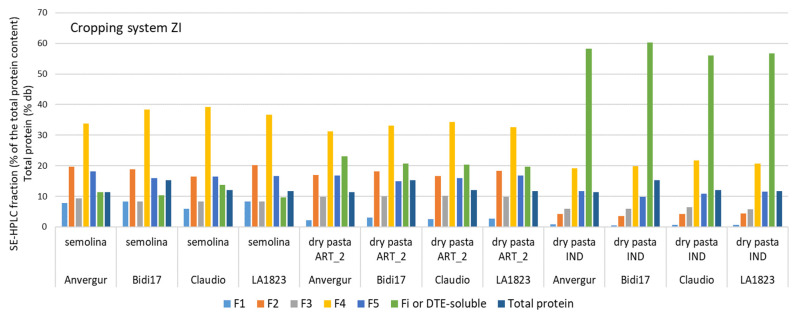
Comparison of the relative proportions of the different protein fractions between stone-milled grains (semolina) grown in 2017 under ZI cropping system and 2 different processes: “Artisanal 2” (ART_2) and “Industrial” (IND).

**Figure 8 foods-12-01093-f008:**
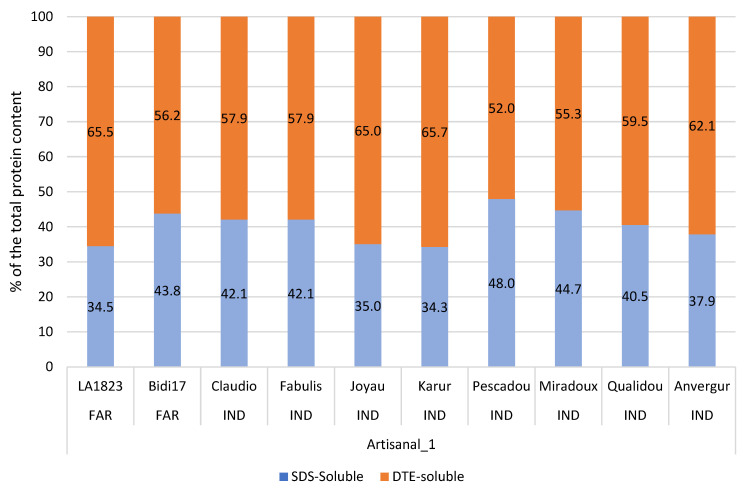
Protein composition measured on freeze-dried cooked pasta from set 1 (H2016 Artisanal 1 process) using SE-HPLC analysis: SDS-soluble and DTE-soluble proteins (as a percentage of the total protein content) according to the cultivar.

**Figure 9 foods-12-01093-f009:**
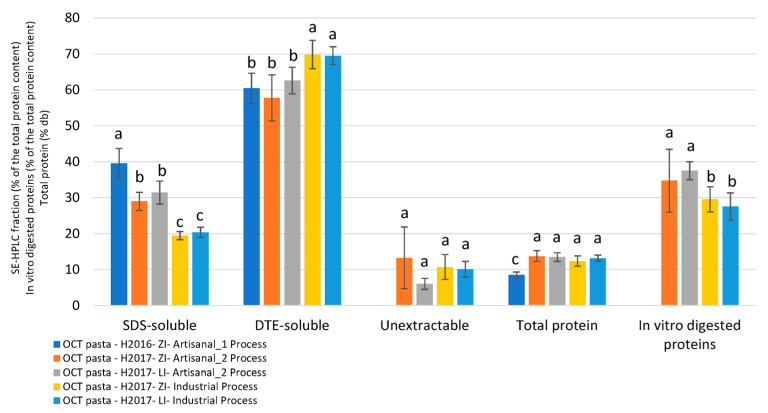
Protein composition measured on freeze-dried OCT cooked pasta from Set 2, using SE-HPLC analysis: SDS-soluble and DTE-soluble proteins (as a percentage of the total protein content), total protein content (% db) and proportion of in vitro-digested proteins (% of the total protein content) according to process and cropping system. Means and error bars (SD). Bar values marked with the same letter are not significantly different based on Student–Newman–Keuls (SNK) test performed at α = 0.05.

**Table 1 foods-12-01093-t001:** Varieties of wheat and number of replicates analyzed by type of product (whole grain, lab-milled grain, stone-milled grain or lab-milled grain, dry and cooked pasta) by variety and over the 3 years of harvest (H2016, H2017, H2019).

Variety	Group	H2016			H2017					H2019	
		Whole Grain	Lab-Milled Grain	Cooked Pasta	Whole Grain	Lab-Milled Grain	Stone-Milled Grain	Dry Pasta	Cooked Pasta	Whole Grain	Lab-Milled Grain
Anvegur	IND	2	3	2	2	4	4	8	8	5	4
Claudio	IND	2	3	2	2	4	4	8	8	5	4
Fabulis	IND	2	3	2	2	4	-	-	-	-	-
Joyau	IND	2	3	2	2	4	-	-	-	-	-
Karur	IND	2	3	2	2	4	-	-	-	-	-
Miradoux	IND	2	3	2	2	4	-	-	-	-	-
Pescadou	IND	2	3	2	2	4	-	-	-	-	-
Qualidou	IND	2	3	2	2	4	-	-	-	-	-
Bidi 17	FAR	2	3	2	2	4	4	8	8	5	5
Bidi 17_2 *	FAR	2	3	-	2	4	-	-	-	-	-
LA1823	FAR	2	3	2	2	4	4	8	8	5	4
LA1823_2 *	FAR	-	3	-	2	-	-	-	-	-	-
Oued Zenati	FAR	-	3	-	2	4	-	-	-	-	-
RG218	FAR	2	3	-	2	4	-	-	-	-	-
Clovis	DIV	2	3	-	2	4	-	-	-	-	-
Desire	DIV	2	3	-	2	4	-	-	-	-	-
Meliani 1B	DIV	2	3	-	2	4	-	-	-	-	-
Vivit	DIV	2	3	-	2	4	-	-	-	-	-
RG1110	DIV	-	3	-	2	4	-	-	-	-	-
RG69649	DIV	2	3	-	2	4	-	-	-	-	-

IND: varieties recommended by industry, FAR: varieties used by farmers, DIV: underutilized genetic resources, -: not available. * sources of the seeds are different.

**Table 2 foods-12-01093-t002:** Pasta making conditions used in “Artisanal 1”, “Artisanal 2” and, “Industrial” processes.

Process	Year of Harvest	Milling	Type of Pasta Press (Manufacturer, Location)	Quantity of Semolina (kg)	Hydration (% db)	Vacuum during Extrusion	Extrusion Temperature (°C)	Drying Step
Artisanal 1	2016	SM 1	P3 (La Montferrina, Montcalieri, Italy)	3	~50	no	Not controlled	Air drying
Artisanal 2	2017	SM 2	Mac 30 (Italpast, Parma, Italy)	3	~50	no	Not controlled	37 °C 48 h
Industrial	2017	SM 2	Experimental mini-press (Sercom, Montpellier, France)	0.7	47	Yes	40 °C	90 °C 5 h

SM: stone milling.

**Table 3 foods-12-01093-t003:** Average of the thousand-kernel weight (TKW), specific weight (SW) and protein content (PROT) measured on durum wheat whole grains by NIRS method, according to the year, the cropping system and the location. Varieties used by farmers (FAR), varieties recommended by industry (IND), other varieties (DIV). Contribution (%) of each factor (variety and environment) to the variability of the criteria TKW, SW and PROT.

Traits	Year of Harvest	H2016	H2017		H2019	
	Cropping System	Zero Input	Zero Input	Low input	Zero Input	Zero Input
	Location	Mauguio	Mauguio	Mauguio	Mauguio	Purpan
	Number of Studied Varieties	18	20	20	4	4
**TKW** **(g)**	**Mean** Min Max	**42.6** 37.9 51.9	**39.8** 33.8 48.5	**48.2** 40.8 57.1	**51.4** 44.5 57.5	**56.6** 52.1 64.2
	FAR	49.6 a ^§^	44.5 a	54.5 a	55.2 a	60.4 a
	IND	40.8 c	38.0 b	44.7 c	47.5 b	52.7 b
	DIV	42.6 b	38.6 b	47.5 b	-	-
	*mean of cv. Bidi17 and cv. LA1823 (FAR)*	*49.4 a*	*45.7 a*	*52.7 a*	*55.2 a*	*60.4 a*
	*mean of cv. Anvergur and cv. Claudio (IND)*	*42.8 b*	*42.6 b*	*48.4 b*	*47.5 b*	*52.7 b*
**SW** **(kg.hL^−1^)**	**Mean** Min Max	**82.5** 80.1 85.7	**80.7** 74.0 86.0	**83.5** 79.6 86.0	**86.7** 85.3 87.6	**84.7** 83.1 86.0
	FAR	85.4 a	83.4 a	84.6 a	87.1 a	85.3 a
	IND	81.7 c	78.8 c	82.3 b	86.4 b	84.2 b
	DIV	83.0 b	81.0 b	84.1 a	-	-
	*mean of cv. Bidi17 and cv. LA1823 (FAR)*	*85.4 a*	*85.0 a*	*85.2 a*	*87.1 a*	*85.3 a*
	*mean of cv. Anvergur and cv. Claudio (IND)*	*82.8 b*	*83.7 b*	*84.6 b*	*86.4 b*	*84.2 b*
**PROT** (% db)	**Mean** Min Max	**9.0** 7.5 10.6	**13.6** 11.2 17.5	**13.6** 12.1 16.3	**11.3** 10.7 12.1	**10.2** 9.1 10.9
	FAR	9.7 a	13.6 b	14.6 a	11.9 a	10.8 a
	IND	8.6 c	12.7 c	12.7 b	10.7 b	9.7 b
	DIV	9.3 b	14.6 a	14.0 a	-	-
	*mean of cv. Bidi17 and cv. LA1823 (FAR)*	*9.4 a*	*13.2 a*	*13.8 a*	*11.9 a*	*10.8 a*
	*mean of cv. Anvergur and cv. Claudio (IND)*	*8.4 b*	*11.5 b*	*13.2 a*	*10.7 b*	*9.7 b*
** *SS Factor/SS total* **	** *TKW* **		** *SW* **		** *PROT* **	
*Variety*	*67%*		*68%*		*42%*	
*Environment*	*29%*		*20%*		*48%*	
*Error*	*4%*		*12%*		*10%*	

^§^ In a same sub-column, numbers followed by the same letter are not significantly different based on Student–Newman–Keuls (SNK) test performed at α = 0.05. -: not available.

**Table 4 foods-12-01093-t004:** Contribution of each factor (Variety, Type, Year, Cropping system, and Location) to the variability of parameters concerning protein composition determined by SE-HPLC. Variance percentages coming from ANOVAs on lab-milled grains from the 2017 harvest (Sum of squares of each factor/total sum of squares) for each of the protein fraction.

		SE-HPLC Protein Composition								Total Protein
		F1 ^†^	F2	F3	F4	F5	Fi	UPP	Glia/Glu	
All years	Variety (*n* = 19)	37% **^,‡^	17% **	10% **	10% **	4% *	14% **	15% **	26% **	15% **
	Type (*n* = 3)	2% ns	5% **	2% *	3% **	1% *	8% **	7% **	6% **	12% **
	Year (*n* = 3)	1% ns	68% **	67% **	80% **	82% **	62% **	64% **	49% **	59% **
	Cropping system (*n* = 2)	2% *	2% **	<1% ns	<1% *	<1% *	3% **	3% **	<1% *	1% **
	Location (*n* = 2)	<1% ns	<1% ns	<1% ns	<1% *	<1% *	<1% *	<1% *	<1% ns	4% **
	Var × CS	11% ns	1% ns	4% ns	3% **	2% ns	5% **	3% **	6% **	3% **
	Var × Location	<1% ns	<1% ns	<1% ns	<1% ns	<1% ns	1% *	1% *	<1% ns	<1% *
	Residual	46%	6%	18%	4%	10%	7%	5%	11%	4%
2017	Variety	67% **	72% **	68% **	75% **	60% **	53% **	55% **	78% **	83% **
	Cropping system	4% *	13% **	1%	2% *	4% *	17% **	19% **	1% *	5% **
	Variety × cropping system	10%	12% **	7%	14% *	15%	24% **	22% **	14% **	10% **
	Residual	19%	3%	24%	9%	21%	6%	4%	7%	2%

^†^ F1 = large glutenin aggregates; F2 = medium and small glutenin aggregates; F3 = ω-gliadins and high-molecular-weight albumins; F4 = α- β- and γ-gliadins; F5 = albumins-globulins, Fi = insoluble fraction; UPP: unextractable polymeric proteins; Glia/Glu: gliadin to glutenin ratio. ^‡^ The contribution is calculated as the sum of squares of the factor/total sum of squares of the model and expressed in %. The model is as follows: Param = Type of variety + Variety effect + Year effect + Cropping system effect + Location effect + (Var × CS) + (Var × location)+ residual. ** Significant at 0.1%, * significant at 5%; ns: not significant.

**Table 5 foods-12-01093-t005:** Means of the protein quality parameters measured on ground grains (lab-milled grains) of cvs Anvergur, Claudio, Bidi17 and LA1823 grown under zero-input conditions and harvested in Mauguio in 2016, 2017 and 2019.

Harvested Year	Total Protein (% db)	SE-HPLC Protein Fraction (%)						UPP (%)	Glia/Glu
		F1	F2	F3	F4	F5	Fi		
2016	8.1 c ^§^	6.9 a	22.7 a	12.4 a	29.5 c	22.3 a	6.1 c	17.1 c	0.83 c
2017	10.2 b	6.6 a	18.1 c	9.2 c	37.0 a	17.8 c	11.3 a	31.6 a	1.03 a
2019	10.8 a	6.8 a	20.0 b	10.0 b	33.7 b	19.7 b	9.6 b	26.4 b	0.92 b
Sum of squares = SS effect/SS model									
Variety	30% **	71% **	33% **	1%	6% **	6%	35% **	37% **	13% **
Year	65% **	6%	64% **	79% **	90% **	83% **	62% **	61% **	78% **
Var × year	4% **	6%	1%	2%	3% *	5%	1%	1%	6% *
Residuals	1%	17%	2%	18%	1%	6%	2%	1%	3%

^§^ Data followed by the same letter are not significantly different at 5% level. ** Significant at 0.1%, * significant at 5%.

**Table 6 foods-12-01093-t006:** Means of the protein quality parameters measured on ground grains (lab-milled grains) grown under zero-input conditions (ZI) and low input conditions (LI) in Mauguio in 2017, and Sum of squares (SS Effect/SS Model) for the variety and location effects. Mean of protein quality criteria measured on ground grains harvested in 2019 in 2 locations (Mauguio and Purpan), and Sum of squares (SS Effect/SS Model) for the variety effect and location effect.

		Total Protein (% db)	SE-HPLC Protein Fraction (%)						UPP %	Glia/Glu
			F1	F2	F3	F4	F5	Fi		
2017	LI	11.3 a ^§^	7.3 a	19.3 a	9.4 a	36.6 a	17.8 b	9.6 b	26.6 b	1.01 a
	ZI	10.8 b	6.9 b	18.5 b	9.2 a	36.1 a	18.3 a	11.1 a	30.3 a	0.99 a
	Sum of squares									
	Variety	**	**	**	**	**	**	**	**	**
	Cropping system	**	*	**	ns	ns	*	**	**	ns
2019	Mauguio	10.8 a ^§^	6.8 b	20.0 a	10.0 b	33.7 a	19.7 b	9.6 a	26.4 a	0.92 a
	Purpan	8.8 b	7.3 a	20.6 a	10.3 a	32.6 b	20.7 a	8.5 a	23.3 a	0.89 b
	Sum of squares									
	Variety	*	**	**	*	*	ns	*	**	**
	Location	**	*	ns	*	*	*	ns	ns	*

^§^ Data followed by the same letter are not significantly different at 5% level. ** Significant at 0.1%, * significant at 5%; ns, not significant.

**Table 7 foods-12-01093-t007:** Means of protein content and protein fractions measured on lab-milled grains (H2016, H2017 and H2019) in 3 groups of varieties. FAR: varieties used by the farmer pasta makers; IND: varieties recommended by the industry; DIV: underutilized genetic resources.

	SE-HPLC Protein Fraction (%)						UPP (%)	Glia/Glu	Total Protein (% db)
	F1	F2	F3	F4	F5	Fi			
FAR	7.4 a ^§^	21.3 a	10.4 b	34.1 a	19.7 b	7.1 c	19.8 c	0.95 a	10.5 a
IND	7.1 a	20.5 b	10.4 b	32.7 c	20.2 a	9.1 a	24.8 a	0.89 b	9.1 c
DIV	7.1 a	20.0 c	10.9 a	33.5 b	20.1 a	8.4 b	23.4 b	0.95 a	10.2 b

^§^ Data followed by the same letter are not significantly different at 5% level.

**Table 8 foods-12-01093-t008:** Contribution of each factor (Sum of squares of the different factors (Variety, Cropping system)/total sum of squares) to the variability of the protein fractions in semolina (stone-milled grains). The model of analysis of variance was: Param = Variety + Cropping system + (variety × cropping system) + residuals, with Param = the protein fraction, UPP or the protein content.

% SS Factor/Total SS	SE-HPLC Protein Composition								Total Protein
	F1	F2	F3	F4	F5	Fi	UPP	Glia/Glu	
Variety	93% **	84% **	46% *	49% *	22% **	68% *	83% **	56% *	86% **
Cropping system	<0.3%	0.3%	<1%	8% *	48% **	<0.1%	<0.1%	3%	1.5% **
Var × CS	4% *	0.7%	30% *	30% *	28% **	13%	4.5%	28% *	12.5% **
Residuals	3%	15%	23%	13%	2%	19%	12.5%	13%	0%

** Significant at 0.01%, * significant at 5%.

**Table 9 foods-12-01093-t009:** Contribution of each factor to the variability of the protein fractions in dry pasta, calculated by the ratio: Sum of squares of the factor/total sum of squares of the model.

	SE-HPLC Protein Composition						
	F1	F2	F3	F4	F5	DTE-Soluble	Unextr.
Variety (*n* = 4)	1%	0.6%	1%	1%	2%	1%	29% *
Cropping system (CS) (*n* = 2)	<<1%	0.4%	2%	1%	2% *	<<1%	6%
Process (*n* = 2)	94% *	98% *	92% *	96% *	88% *	98% *	49% *
Residual of the model (including the interactions)	4%	1%	5%	2%	8% (Var × CS = 6% *)	1%	16% (Var × Process = 7% *)

* Significant at 5%.

**Table 10 foods-12-01093-t010:** Correlations between the protein fractions analyzed on ground grains and those analyzed on semolina and dry pasta.

Lab-Milled Grains	SE-HPLC Protein Composition					
	F1	F2	F3	F4	F5	Fi or DTE-Soluble
with Stone-milled grains	0.89 **	0.93 **	0.15 ns	0.89 **	0.66 ns	0.55 ns
with Dry Pasta	0.06 ns	0.06 ns	0.05 ns	0.11 ns	0.23 ns	0.05 ns

** Significant at 0.1%; ns: not significant.

**Table 11 foods-12-01093-t011:** Contribution of each factor to the variability of the parameters: percentage of proteins soluble in the SDS, proteins soluble in the DTE, unextractable proteins, and total protein content.

%SS Factor/SS Tot	SE-HPLC Protein Composition			Total Protein
	SDS-Soluble	DTE-Soluble	Unextractable Proteins	
Variety	4% *	16% *	34% *	56% **
Cropping system	1%	1%	4.5%	4% *
Process	84% **	53% **	0.5%	12% **
Residual of the model (including the significant interactions)	11% (var × process = 3% * var × CS = 4% *)	30% (var × process = 5.5% * var × CS = 11% *)	61% (var × process = 17.5% *)	28% (var × process = 5% * var × CS = 14% ** process × CS = 2% *)

** Significant at 0.1%; * significant at 5%; ns, not significant.

## Data Availability

In the frame of open science encouraged by INRAE, the raw data will be available as soon as possible on https://entrepot.recherche.data.gouv.fr/dataverse/inrae.

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
