# Peer review of "Farmer vs. Industrial Practices: Impact of Variety, Cropping System and Process on the Quality of Durum Wheat Grains and Final Products"

_foods, 2023, doi:10.3390/foods12051093_

Round 1
Reviewer 1 Report
The paper in general is well written, the experimental design is straightforward, the results are clear. Although genetic and environmental variability of protein content and composition has been a known issue for quite some time, the comparison on industrial and artisanal practices adds novelty to the study. There is one aspect that I consider a big problem though. The authors bring in gluten sensitivites very often, although durum wheat is not known to be applicable in the diet of these patients, the supporting evidence they bring up is mostly anecdotal and they did not perform any immunological studies whatsoever to give substance to this side of the paper. In my opinion any mention of gluten sensitivites should be entirely removed and the paper should only focus on the comparative study without this aspect, until this happens it is pointless to go into a more detailed assessment.
Author Response
Response of the authors to the remarks of the reviewer 1
The authors want, firstly, to warmly thank the reviewer 1 for his/her very attentive reading of the document, for his/her encouragement and for the relevance of these remarks.
We detail hereafter, point by point, the way we took into account each of the mentioned points.
Rev1 : (x) English language and style are fine/minor spell check required
Resp. of authors : The entire manuscript has been revised to improve the style and eliminate grammatical errors.
Rev 1 : The paper in general is well written, the experimental design is straightforward, the results are clear. Although genetic and environmental variability of protein content and composition has been a known issue for quite some time, the comparison on industrial and artisanal practices adds novelty to the study. There is one aspect that I consider a big problem though. The authors bring in gluten sensitivites very often, although durum wheat is not known to be applicable in the diet of these patients, the supporting evidence they bring up is mostly anecdotal and they did not perform any immunological studies whatsoever to give substance to this side of the paper. In my opinion any mention of gluten sensitivites should be entirely removed and the paper should only focus on the comparative study without this aspect, until this happens it is pointless to go into a more detailed assessment.
Resp of authors : The Reviewer 1 point out a very relevant issue. The project originates with the statements of people suffering from NCGS (Non Coeliac Gluten Sensitivity) and claiming that they can consume any artisanal cereal products while industrial products are harmful to their health. For this reason, we decided to study what differentiates peasant products and industrial products. But it is true that our results do not report any immunological or clinical studies on the reaction of NCGS people to the ingestion of such products. Following your remark, we have therefore rewrite the introduction (L33- L142) in order to set the framework more clearly, and removed most of the NCGS mentions.
To conclude, the authors again thank reviewer 1 for the constructive remarks that allowed them to improve the understanding and quality of the manuscript.
Reviewer 2 Report
The authors present extensive experimental work on the protein fraction of durum wheat varieties, aimed at evaluating any correspondences with the greater acceptability of pasta products obtained by non-celiac subjects but affected by NCGS (Non-Celiac Gluten Sensitivity) diseases.
Due to its primary objective, the paper appears to be more aimed at an audience of nutritionists, and therefore could find a more appropriate placement in other journals of the group. However, the work fits adequately to the journal's scope since it also includes an evaluation of the technological process of obtaining pasta, an equally central objective in the experimental plan presented by the authors.
The study of the bibliography shows a considerable effort to identify the most interesting literature papers for the subject, both from a varietal, agronomic and technological point of view. The study is reported on various years of production and is somewhat difficult to follow in some respects. However, the exposition of the materials and methods appears rigorous and precise.
In the results section, there is a need to better clarify some methodologies, such as a better explanation (in the paragraph or in materials and methods) of the NIRS method mentioned in paragraph 3.1.3., relating to the protein content.
The results, however, do not lead to a sufficiently exhaustive answer on the interrelationship between the use for pasta production of artisanal varieties and the lower impact on NCGS pathologies. Therefore, it appears necessary to reformulate the paragraph of the conclusions highlighting more clearly the effective scope of the results obtained, and referring to further research to evaluate the role of the higher protein content found in the artisanal varieties.
In this reviewer's opinion, if in the first place, the editor deems the submission appropriate for the purposes of the journal, it is considered appropriate to make minor changes to the text to underline the actual results obtained. Furthermore, it would be advisable to reformulate the headlines of paragraphs 4.2, 4.4 and 4.5 by eliminating the interrogative form.
Author Response
Response of the authors to the remarks of the reviewer 2
The authors thank the reviewer 2 for the relevance of these remarks due to his/her very attentive reading of the document.
We detail hereafter, point by point, the way we took into account each of the mentioned points.
Rev 2 : (x) English language and style are fine/minor spell check required
Resp. of authors : The entire manuscript has been revised to improve the style and eliminate grammatical errors.
Rev 2 : “The authors present extensive experimental work on the protein fraction of durum wheat varieties, aimed at evaluating any correspondences with the greater acceptability of pasta products obtained by non-celiac subjects but affected by NCGS (Non-Celiac Gluten Sensitivity) diseases.
Due to its primary objective, the paper appears to be more aimed at an audience of nutritionists, and therefore could find a more appropriate placement in other journals of the group. However, the work fits adequately to the journal's scope since it also includes an evaluation of the technological process of obtaining pasta, an equally central objective in the experimental plan presented by the authors.”
Resp. of authors : The authors appreciate the understanding of Reviewer 2. Indeed the origin of the project is the observation made by people declaring themselves NCGS and who can nevertheless consume artisanal cereal products without experiencing the gastric discomfort occurring after ingestion of industrial cereal products. This is the origin but not the scope of the paper, as reviewer 2 well understood. We modified the introduction (lines 33-142) in order to be clearer on this point.
Rev2 : “The study of the bibliography shows a considerable effort to identify the most interesting literature papers for the subject, both from a varietal, agronomic and technological point of view. The study is reported on various years of production and is somewhat difficult to follow in some respects. However, the exposition of the materials and methods appears rigorous and precise.”
Resp. of authors : We have taken note of this remark and have modified the text (Table 1) to clarify this point
Rev 2 : “In the results section, there is a need to better clarify some methodologies, such as a better explanation (in the paragraph or in materials and methods) of the NIRS method mentioned in paragraph 3.1.3., relating to the protein content.”
Resp. of authors : We added some explanation concerning NIRS method (lines 169-173)
Rev 2 : “The results, however, do not lead to a sufficiently exhaustive answer on the interrelationship between the use for pasta production of artisanal varieties and the lower impact on NCGS pathologies. Therefore, it appears necessary to reformulate the paragraph of the conclusions highlighting more clearly the effective scope of the results obtained, and referring to further research to evaluate the role of the higher protein content found in the artisanal varieties.”
Resp. of authors: The aim of this paper is not to show the relation between artisanal products and NCGS. The experiments have not been implemented in this way. But to take into account your remark, we have reformulated the introduction (L33-L142) to better explain the objectives.
Rev. 2 : “In this reviewer's opinion, if in the first place, the editor deems the submission appropriate for the purposes of the journal, it is considered appropriate to make minor changes to the text to underline the actual results obtained. Furthermore, it would be advisable to reformulate the headlines of paragraphs 4.2, 4.4 and 4.5 by eliminating the interrogative form.”
Resp. of authors: We have eliminated the interrogative form for the headlines of the mentioned paragraphs (Lines 703, 806, 844)
Reviewer 3 Report
I have compiled all comments and assessments on the manuscript as follows:
Review
Samson et al. „Farmer vs industrial practices: Impact of variety, cropping system and process on the quality of durum wheat grains and final 3 products (semolina and pasta)”
In detail, the reviewer has the following comments on the text:
L = Line
L81-83: Here, in the middle of the introduction, the objectives of the investigations carried out are stated. However, this is followed by about one and a half pages of further introductions. And at the end, it tells what was done. It is therefore recommended to structure the introduction better and to state the objectives and hypotheses at the end.
L165 and following: The reason why the trial year 2018 was not included is missing. Furthermore, it should be stated whether the varieties were winter or summer forms.
L166: Italic spelling for botanical names.
L178: Were no fungicides applied? What does "150u N" mean? What does the "u" mean? Was the 150 N applied in one dose or in splitting? Information about the Nmin content (mineral N) in the soil is missing.
L 272 Remarks on statistics:
Different indications of significance or probability of error are given in the results section. For example: significant at 5% or 1%. This probably means the probability of error "alpha", i.e., alpha=5% or 1%. If, on the other hand, one is talking about p-values, one should write p = 0.05 (=alpha 5%) or p = 0.01 (=alpha 1%). This should be clarified in the chapter "Statistics" and handled consistently throughout the manuscript.
L276: "SNK" should be explained here as "Student-Newman-Keuls".
L 361: In the header of table 5 and in the following tables the F-values should be explained. In a line above, one could write: "Protein fractions". See also table 8 and further tables.
L412: Figure 3: What is the unit of measurement of the y-axis?
L 423: The title of this chapter does not quite match the content. In addition to the "location effect", Figure 4 also shows variety effects for the year 2016, related to F4 and Fi.
L 446: The abbreviations "FAP" and "IND" should not be used alone in the title. Better would be: Comparison of farmer varieties (FAP) with industry recommended varieties (IND).
L 452: Why is the symbol "§" used here? This should be consistent for all tables.
L 457: There are no statistics in Figure 5, which is why the statements from it can only be described as tendential. Therefore, one should not speak of "significant". Why are other differences (in UPP, Fi ...) not mentioned in the text? Since no statistics were made, the graph can also be deleted.
L 486: This graph is insignificant and should be deleted. This statement can be mentioned in one sentence in the text.
L 537: Figures 5, 6, 8 and 9: Here it was obviously not possible to make a statistical calculation because of too few individual values. It would therefore be useful to know from how many individual values the mean values were formed here. Why was the standard deviation of the standard error per mean not calculated?
L 587: It would be good to mark the varieties in the figure that belong to the FAR or IND group.
L 626: Calculation of the standard error (SE) per mean would be useful.
L 635: At the end of the results section a result on the enzymatic digestibility of the pasta is mentioned (in-vitro test). Tables or graphs with the data are not presented. For this reason, the statements at this point do not seem very convincing.
L 658 and following: In the first section of the discussion, the results are more or less repeated, and no interpretation is made. Only at the end of chapter 4.1 is an evaluation of the own data made. Therefore, this chapter can be shortened by that part where only observations are repeated.
L 875: The conclusions drawn from the research are pertinent (and acceptable) and provide a glimpse of future research that should be conducted on pasta production.
L 927: The extensive literature sources cited are scientifically relevant and correctly reproduced.
Form and style:
The authors have written the manuscript in reasonably good English. Nevertheless, there are some spelling mistakes in the text (capitalisation, expression, spacing between words) that should be corrected. Overall, however, the writing is good.
Overall assessment
The authors have evaluated a very extensive data material and written a longer manuscript on this basis. The topicality of the subject matter is very high. It is considered positive that the authors have made a comparison between so-called old and new durum varieties with regard to their grain and product properties. Of particular scientific interest are the results on the structure of the proteins (gluten proteins), which were investigated here in relation to the varieties, the environments and the processing. However, a major problem of this study is that not all investigations were carried out in all trial years and with all varieties (and environments). As a result, some comparisons are limited, not all data have been statistically analysed and the representativeness of the statements is somewhat limited. Nevertheless, I consider the results significant and recommend the publications after revising the manuscript.
Author Response
Response of the authors to the remarks of the reviewer 3
The authors greatly appreciated the particularly meticulous and detailed proofreading of reviewer 3 and thank him/her warmly!
They detail hereafter, point by point, the way they took into account each of the mentioned points.
Rev3 : (x) English language and style are fine/minor spell check required
Resp. of authors : The entire manuscript has been revised to improve the style and eliminate grammatical errors.
Rev. 3 : “L81-83: Here, in the middle of the introduction, the objectives of the investigations carried out are stated. However, this is followed by about one and a half pages of further introductions. And at the end, it tells what was done. It is therefore recommended to structure the introduction better and to state the objectives and hypotheses at the end. “
Resp. of authors: the introduction (L33-L142) was modified and shortened, to better clarify the objectives of the paper.
Rev 3 : “L165 and following: The reason why the trial year 2018 was not included is missing. Furthermore, it should be stated whether the varieties were winter or summer forms. “
Resp. of authors : we added information about these points (line 147-148)
Rev 3 : “L166: Italic spelling for botanical names” .:
Resp. of authors : we checked that all the botanical names are now in italic in the entire manuscript.
Rev 3 : “L178: Were no fungicides applied? What does "150u N" mean? What does the "u" mean? Was the 150 N applied in one dose or in splitting? Information about the Nmin content (mineral N) in the soil is missing.”
Resp. of authors : As required here, we added informations concerning agronomical aspects (lines 156-160)
Rev 3 : “ L 272 Remarks on statistics: Different indications of significance or probability of error are given in the results section. For example: significant at 5% or 1%. This probably means the probability of error "alpha", i.e., alpha=5% or 1%. If, on the other hand, one is talking about p-values, one should write p = 0.05 (=alpha 5%) or p = 0.01 (=alpha 1%). This should be clarified in the chapter "Statistics" and handled consistently throughout the manuscript.”
Resp. of authors : we clarified these notions in the chapter “statistics” and we homogenized throughout the manuscript
Rev 3 : “L276: "SNK" should be explained here as "Student-Newman-Keuls".
Resp. of authors : we modified as required (line 267-268).
Rev 3 : “L 361: In the header of table 5 and in the following tables the F-values should be explained. In a line above, one could write: "Protein fractions". See also table 8 and further tables.”
Resp. of authors: The modifications have been done for all tables.
Rev 3 “L412: Figure 3: What is the unit of measurement of the y-axis?”
Resp. of authors: we added the unit on this figure 3.
Rev 3 : “L 423: The title of this chapter does not quite match the content. In addition to the "location effect", Figure 4 also shows variety effects for the year 2016, related to F4 and Fi.”
Resp. of authors : We changed the title and sub-title (line 431).
Rev 3 “L 446: The abbreviations "FAR" and "IND" should not be used alone in the title. Better would be: Comparison of farmer varieties (FAR) with industry recommended varieties (IND).”
Resp. of authors : We changed as suggested here.
Rev 3 : “ L 452: Why is the symbol "§" used here? This should be consistent for all tables”.
Resp. of authors : We used this symbol "§" to differentiate from the other used symbol (*), that are used for the level of significance in the others tables.
Rev 3 : “ L 457: There are no statistics in Figure 5, which is why the statements from it can only be described as tendential. Therefore, one should not speak of "significant". Why are other differences (in UPP, Fi ...) not mentioned in the text? Since no statistics were made, the graph can also be deleted.”
Resp. of authors : : The statistics have been made for all the data but not always presented in the figures. We corrected the figure 5 and we add the error bar representing standard deviation.
Rev 3 “L 486: This graph is insignificant and should be deleted. This statement can be mentioned in one sentence in the text.”
Resp. of authors : We agree with you and we deleted this graph
Rev 3 “L 537: Figures 5, 6, 8 and 9: Here it was obviously not possible to make a statistical calculation because of too few individual values. It would therefore be useful to know from how many individual values the mean values were formed here. Why was the standard deviation of the standard error per mean not calculated?”
Resp. of authors : : We realized statistical analysis on the elementary data and presented on the figures only the means. Based on your remarks, we combined several figures and added on the figures the std data.
Rev 3 “L 587: It would be good to mark the varieties in the figure that belong to the FAR or IND group.”
Resp. of authors : We corrected that by marking the varieties. (figure 10, line 587)
Rev 3 “L 626: Calculation of the standard error (SE) per mean would be useful.”
Resp. of authors : We added error bars (std) and letters
Rev 3 “L 635: At the end of the results section a result on the enzymatic digestibility of the pasta is mentioned (in-vitro test). Tables or graphs with the data are not presented. For this reason, the statements at this point do not seem very convincing.”
Resp.of authors The date are presented in the supplementary data (table S5)
Rev 3 “L 658 and following: In the first section of the discussion, the results are more or less repeated, and no interpretation is made. Only at the end of chapter 4.1 is an evaluation of the own data made. Therefore, this chapter can be shortened by that part where only observations are repeated.”
Resp.of authors : we rewritten this section.
Rev 3 “L 875: The conclusions drawn from the research are pertinent (and acceptable) and provide a glimpse of future research that should be conducted on pasta production.”
Resp.of authors : Thanks for this encouragement !
Rev 3 “ L 927: The extensive literature sources cited are scientifically relevant and correctly reproduced”.
Resp.of authors : Thanks !
Rev 3 : “Overall assessment : The authors have evaluated a very extensive data material and written a longer manuscript on this basis. The topicality of the subject matter is very high. It is considered positive that the authors have made a comparison between so-called old and new durum varieties with regard to their grain and product properties. Of particular scientific interest are the results on the structure of the proteins (gluten proteins), which were investigated here in relation to the varieties, the environments and the processing. However, a major problem of this study is that not all investigations were carried out in all trial years and with all varieties (and environments). As a result, some comparisons are limited, not all data have been statistically analysed and the representativeness of the statements is somewhat limited. Nevertheless, I consider the results significant and recommend the publications after revising the manuscript.”
Resp.of authors: To conclude, the authors sincerely thank the reviewer 3 for all these remarks which have led to a substantial improvement of the manuscript
Reviewer 4 Report
This study investigated the combined impact of durum wheat varieties, cropping systems, and processes on the quantity and quality of gluten and its digestibility. The idea of the paper is of interest. However, the introduction section, the way results have been presented and discussed need to be carefully revised. Although the results are direct and straightforward, authors tended to complicate them with too many tables and figures. I think the paper should be carefully revised throughout.
I have made my comments directly into the MS. These comments and editing did not cover the entire MS but rather are examples to what can be done for the rest of the paper.

Reviewer 5 Report
foods-2196363-peer-review-v1
The paper: Farmer vs industrial practices: Impact of variety, cropping system and process on the quality of durum wheat grains and final 3 products (semolina and pasta). The manuscript idea is great, the parts of the manuscript were narrated in a wonderful and interesting way.
Title: the title of the manuscript is appropriate and attractive.
Abstract: good section
Introduction:
The paragraphs of the introduction must be rewritten and arranged. For example, the aim of the research is written at the end of the introduction. The paragraph on gluten is unnecessary, and the introduction, in general, can be shortened, with a focus on Farmer vs industrial practices: Impact of variety, cropping system, and process on the quality of durum wheat grains and final products (semolina and pasta).
Materials and Methods:
The section on the design of experiments (175-181) is not clear, how the experiment was designed with a split block system.
Why did he consider the different years as a factor for the study?
Results and Discussion
Paragraphs (288-283) must be moved to the materials and methods section
Figure (1) is of very poor quality.
In Figures 2, 5, 6 ,8 ,9 ,10 and 11, some information is missing (Axis Y).
Line 694: Aktan & Khan (1992)[33] or De Zorzi et al. (2007)[52], remove (1992) and (2007).
Line 696: Wagner et al. (2011)[53], remove (2011).
Line 699: Martin et al. (2019)[54], remove (2019).
Lines 745-750, 759, 763, 771, 775, 822, 833 and 853:874: references publication years should be removed.
Line 839: add the reference number.
Conclusions: Excellent.
Best Regards
Author Response
Response of the authors to the remarks of the reviewer 5
The authors sincerely thank the reviewer 5 for his/her very encouraging comments, for his/her careful reading of the manuscript, which gave rise to relevant remarks and led to improving the quality of the article.
Rev 5 : “The paper: Farmer vs industrial practices: Impact of variety, cropping system and process on the quality of durum wheat grains and final 3 products (semolina and pasta). The manuscript idea is great, the parts of the manuscript were narrated in a wonderful and interesting way. Title: the title of the manuscript is appropriate and attractive. Abstract: good section. »
Resp.of authors : Thanks a lot for your encouragement !
Rev 5 : « Introduction: The paragraphs of the introduction must be rewritten and arranged. For example, the aim of the research is written at the end of the introduction. The paragraph on gluten is unnecessary, and the introduction, in general, can be shortened, with a focus on Farmer vs industrial practices: Impact of variety, cropping system, and process on the quality of durum wheat grains and final products (semolina and pasta).”
Resp.of authors : we rewritten and shortened the introduction in order to take into account your remarks (lines 33 to 142).
Rev 5 “Materials and Methods: The section on the design of experiments (175-181) is not clear, how the experiment was designed with a split block system.”
Resp.of authors : we added an explanation concerning this point (lines 154-161)
Rev. 5 “Why did he consider the different years as a factor for the study?”
Resp.of authors : We considered the different years as a factor just to appreciate and rank the impact of the year compared to the others factors (variety, location, process)
Rev.5 : “Results and Discussion - Paragraphs (288-283) must be moved to the materials and methods section”
Resp.of authors : A part of this paragraph was moved and rewrited and the table 4 was removed.
Rev 5 : “Figure (1) is of very poor quality.”
Resp.of authors : We copy/paste a higher quality for this figure.
Rev.5 : “In Figures 2, 5, 6 ,8 ,9 ,10 and 11, some information is missing (Axis Y).”
Resp.of authors : Informations concerning Axis – Y on the different figures was added.
Rev. 5 : “Line 694: Aktan & Khan (1992)[33] or De Zorzi et al. (2007)[52], remove (1992) and (2007)., Line 696: Wagner et al. (2011)[53], remove (2011)., Line 699: Martin et al. (2019)[54], remove (2019)., Lines 745-750, 759, 763, 771, 775, 822, 833 and 853:874: references publication years should be removed., Line 839: add the reference number.”
Resp.of authors : we took into account all of these corrections
Rev 5 : “Conclusions: Excellent”.
Resp.of authors : Thanks a lot !
Round 2
Reviewer 1 Report
Although the paper improved in terms of discussing gluten related disorders, I still feel it still swings to the deceptive side. The authors removed any mention of celiac disease, which is good. But the paper they cite as the basis of their study is based on self-reported patient data and that always must be handled with care. The cited paper appears to be a single-page document in French citing 31 self-reported cases of gluten sensitivity from 10+ years ago. Did these people have a formal medical assessment and a diagnosis signed by a medical professional? Do we need to accept the hypothesis of durum wheat being safe in these diseases just by people feeling so? Even the authors state in the paper that no wheat component could be undoubtedly made responsible for NCGS. I do feel that this part is standing on very weak legs. I still recommend removing every single mention of this aspect. The paper will still stand. And later, when the authors performed more experiments specifically about the gluten sensitivity issues, as a separate study. It is not a lot of change at this point and the paper could still be acceptable on its own merit just by comparing these two methods. But dragging in gluten sensitivities at this point does reduce its value by suggesting things that are not proven by their results and may confer a message that could be harmful for patients actually suffering from these diseases.
Author Response
Rev.1 : Although the paper improved in terms of discussing gluten related disorders, I still feel it still swings to the deceptive side. The authors removed any mention of celiac disease, which is good. But the paper they cite as the basis of their study is based on self-reported patient data and that always must be handled with care. The cited paper appears to be a single-page document in French citing 31 self-reported cases of gluten sensitivity from 10+ years ago.
Authors : We introduced a new reference because the data of the survey are now published : [3] Grégori Akermann, Paul Coeurquetin. Consommer sans gluten : trajectoires individuelles d’éviction. Goulet, Frédéric; Vinck, Dominique. Faire sans, faire avec moins : les nouveaux horizons de l’innovation, Presses des Mines, 2022, Sciences sociales, 978-2-35671-763-4. ⟨hal-03575407⟩⟨hal-03575407⟩
Rev.1 : Do we need to accept the hypothesis of durum wheat being safe in these diseases just by people feeling so?
Authors : We do not at all affirm that durum wheat is safer, but we question the differences between artisanal and industrial products on the basis of the feelings of numerous consumers suffering from gastric disorders (pers. comm or survey data). We focused on gluten proteins considered as the main putative causal agent.
Rev1 : Even the authors state in the paper that no wheat component could be undoubtedly made responsible for NCGS. I do feel that this part is standing on very weak legs. I still recommend removing every single mention of this aspect. The paper will still stand. But dragging in gluten sensitivities at this point does reduce its value by suggesting things that are not proven by their results and may confer a message that could be harmful for patients actually suffering from these diseases.
Authors : The abstract and the discussion were revised in order to remove almost all the mentions to NCGS. We leave only one mention because the initial project came from the observation made by NCGS people and pasta-makers farmers. We set up this project in connection with the French association of celiac patients and the discussions during the project led us to be cautious about the conclusions and the links with the disease.
Reviewer 4 Report
Dear Authors:
Although you have revised your MS and significantly improved the quality of the paper, I think you can further improve the quality of your paper by considering the way you presented your valuable results and the style of your writing-up.
I again made my comments directly to the manuscript as can be seen in the attached file.

Reviewer 5 Report
I have no additional comments.
Author Response
The authors warmly thank reviewer 5 for his/her careful second proofreading